# PortLLM: Personalizing Evolving Large Language Models with Training-Free and Portable Model Patches

**Rana Muhammad Shahroz Khan**[1]    **Pingzhi Li**[1*]    **Sukwon Yun**[1*]    **Zhenyu Wang**[1]
**Shahriar Nirjon**[1]    **Chau-Wai Wong**[2]    **Tianlong Chen**[1]
[1]The University of North Carolina at Chapel Hill    [2]NC State University
* Denotes Equal Contribution

## Abstract

As large language models (LLMs) increasingly shape the AI landscape, fine-tuning pretrained models has become more popular than it was in the pre-LLM era for achieving optimal performance in domain-specific tasks. However, pretrained LLMs such as ChatGPT are periodically evolved (*i.e.*, model parameters are frequently updated), making it challenging for downstream users with limited resources to keep up with fine-tuning the newest LLMs for their domain application. Even though fine-tuning costs have nowadays been reduced thanks to the innovations in parameter-efficient fine-tuning such as low-rank adaptation (LoRA), not all downstream users have adequate computing for *frequent personalization*. Moreover, access to fine-tuning datasets, particularly in sensitive domains such as healthcare, can be time-restrictive, making it crucial to retain the knowledge encoded in earlier fine-tuned rounds for future adaptation. In this paper, we present PortLLM, a training-free framework that (*i*) creates an initial lightweight model update patch to capture domain-specific knowledge, and (*ii*) allows a subsequent seamless plugging for the continual personalization of the evolved LLM at minimal cost. Our extensive experiments cover **seven** representative datasets, from easier question-answering tasks {BoolQ, SST2} to harder reasoning tasks {WinoGrande, GSM8K}, and models including {`Mistral-7B`, `Llama2`, `Llama3.1`, and `Gemma2`}, validating the portability of our designed model patches and showcasing the effectiveness of our proposed framework. For instance, PortLLM achieves comparable performance to LoRA fine-tuning with reductions of up to $12.2\times$ in GPU memory usage. Finally, we provide theoretical justifications to understand the portability of our model update patches, which offers new insights into the theoretical dimension of LLMs' personalization.

## 1 Introduction

The rise of pretrained large language models (LLMs) has marked a significant paradigm shift in natural language processing (NLP), particularly in their ability to adapt to specific domains and tasks. These models, such as GPT-4 (Achiam et al., 2023), have achieved state-of-the-art performance by leveraging vast amounts of pretraining data (Antoniades et al., 2024). However, pretrained LLMs often require adaptation for specialized domains where context-specific knowledge is critical (Wang et al., 2022a; Bommasani et al., 2021; Qiu et al., 2020). Hence, while pretraining provides a strong foundation, *fine-tuning* (*e.g.*, *personalization*) is essential for specific domains. Fine-tuning bridges this gap by adapting pretrained models to specific tasks, enhancing their performance in domains such as healthcare, legal analysis, or scientific research (Min et al., 2021; Wei et al., 2021; Ouyang et al., 2022; Wang et al., 2022b; Liu et al., 2022; Raffel et al., 2020; Chen et al., 2024; Gao et al., 2024b). For instance, fine-tuned models can more effectively recognize domain-specific terminology, reason about complex relationships, and deliver more accurate and contextual appropriate responses.

Much effort has been devoted to developing fine-tuning methods. Traditionally, fine-tuning would normally involve updating all the parameters of a model. For example, in the case of Mistral

7B (Jiang et al., 2023), it would involve updating all 7 billion parameters. Typically, LLMs have billions of parameters, and so this process poses significant challenges in computational and memory requirements. To alleviate these constraints, many parameter efficient fine-tuning (PEFT) methods (Houlsby et al., 2019) have been proposed. One popular method is low-rank adaptation (LoRA) (Hu et al., 2021), which aims to estimate an update matrix $\Delta W$ using the product of two low-rank matrices $A$ and $B$. However, although LoRA lowers the training complexity, it still requires fine-tuning a large number of trainable parameters to reach a satisfactory performance. For example, LoRA fine-tuning a Llama 2 13B (Touvron et al., 2023) variant would require up to eight A6000 GPUs with 48 GB VRAM each, for a very small batch size, hence imposing a considerable memory and computational burden.

Furthermore, as cloud-hosted LLMs like ChatGPT (OpenAI, 2022) and Gemini (Team et al., 2023) undergo periodic (biannual or more frequent) updates with newer data, the best performing model often renders previous versions outdated. For downstream users who have already invested in fine-tuning the previous models for domain-specific tasks, repeatedly fine-tuning or performing personalization at every new update is highly impractical, as this process is not only computationally expensive but also time-consuming.

Beyond the computational and logistical hurdles, continual updates present another challenge for the downstream user: the lack of availability of the fine-tuning dataset. In domains such as healthcare or finance, data access is regulated by privacy laws and potentially time-sensitive. For example, fine-tuning medical models on patient data requires strict adherence to ethical and legal guidelines, making it difficult to repeatedly fine-tune with every LLM release. As a result, repeatedly fine-tuning models on newer LLM releases is not a viable long-term strategy for many users, hindering the downstream users from harnessing the performance gains from the evolving nature of LLMs. In response to these challenges, a natural research question arises:

> ***(Q)** How to leverage the personalized knowledge captured in the first fine-tuned LLM model to update any evolving LLMs?*

To address this question, we introduce PORTLLM, a training-free framework that enables seamless transfer of domain-specific knowledge across evolving models. PORTLLM leverages model patches derived from LoRA, allowing users to port fine-tuned knowledge from one model iteration to another while preserving or even enhancing the performance of a downstream task. We show that our model patches are portable across model updates. If the downstream user fine-tuned a version of the model that has long become obsolete, they can simply add the model patch to the newer model, maintaining or boosting their performance on the downstream task. PORTLLM eliminates the need for costly periodic fine-tuning, offering a scalable solution for maintaining task-specific performance across different model versions. Our contributions can be summarized as follows:

❶ We introduce PORTLLM, a training-free framework designed to transfer knowledge between different versions of an evolving LLM. Given two LLM versions, PORTLLM leverages task-specific model patches extracted from a fine-tuned LLM and seamlessly applies them to the evolved LLM. This process allows the updated LLM to achieve comparable, and in some cases improved, performance on downstream tasks—without any need for fine-tuning.

❷ *Why do our model patches work?* We address this question through both theoretical analysis and empirical investigation. Our findings reveal that certain terms in the model patch are effectively negligible, enabling us to create a simplified version of the patch that requires no training. Consequently, adding the simplified model patches alone is sufficient to achieve improved performance on the downstream task. Furthermore, we examine the impact of the pretraining dataset on downstream tasks, demonstrating that our framework can harness the benefits of continued pretraining across different model updates and across different pretraining datasets including {OpenOrca, SlimOrca, OpenPlatypus, AlpacaGPT4}.

❸ We conduct extensive experiments across a series of **seven** downstream tasks, including Question-Answering Tasks {BoolQ, SST-2} (Wang, 2018; Wang et al., 2019), Similarity and Paraphrase Tasks {MRPC} (Wang, 2018), Inference Tasks {RTE, WNLI} (Wang, 2018), and Reasoning Tasks {WinoGrande, GSM8K} (Sakaguchi et al., 2021; Cobbe et al., 2021). To further demonstrate the robustness and broad applicability of our approach, we evaluate it on multiple model architectures, such as `Mistral-7B`, `Llama2-7B`, `Llama3.1-8B`, and `Gemma2-9B`. Additionally, we explore the applicability of our method on full-weight continued pretraining

compared to LoRA-based continued pretraining and show the effectiveness of our method by quantifying the gains on GPU memory usage (of up to $12.2\times$ reduction), GPU hours, and decrease in the number of trainable parameters (to zero trainable parameters).

## 2 RELATED WORKS

**Large Language Models (LLMs).** LLMs have transformed natural language processing, enabling models to perform complex tasks with remarkable accuracy and generalization. Models like GPT-3 (Brown et al., 2020), BERT (Devlin et al., 2019), and T5 (Raffel et al., 2023) have set benchmarks across a range of NLP tasks, from translation and summarization to question answering and text generation (Vaswani et al., 2023; Zhang et al., 2020; Rajpurkar et al., 2016). More recently, models like Llama (Touvron et al., 2023; Dubey et al., 2024), Mistral (Jiang et al., 2023), and Gemma (Team et al., 2024) have pushed the boundaries further by optimizing both performance and computational efficiency. LLama, Mistral, and Gemma represent recent advances in LLM architectures, each offering improvements in efficiency and performance. However, even with such improvements, the performance on domain-specific downstream tasks is subpar, making fine-tuning necessary. In this paper, we propose a training-free solution that enables the seamless transfer of personalized knowledge across evolving LLMs, reducing the need for costly fine-tuning and enhancing accessibility.

**Parameter Efficient Fine-Tuning (PEFT).** The rapid growth in the size of pretrained LLMs has posed significant challenges for efficiently fine-tuning LLMs to specific downstream tasks. To address this challenge, numerous PEFT methods have been developed, aiming to balance efficiency and accuracy. Early approaches focused on inserting trainable adapters—feed-forward networks placed between layers of the pretrained model (Houlsby et al., 2019; Lin et al., 2020). Recent advancements have led to more sophisticated adapter-based PEFT methods (Mahabadi et al., 2021; Pfeiffer et al., 2020; Luo et al., 2023) including LMaaS (Sun et al., 2022) for service-oriented adaptation and kNN-Adapter (Huang et al., 2023) for retrieval-augmented fine-tuning. A notable example is LoRA (Hu et al., 2021), which introduces trainable low-rank weight perturbations to the pretrained model, significantly reducing the number of parameters required for fine-tuning. LoRA's key innovation lies in its use of the product of two low-rank matrices to approximate weight changes. Building upon this concept, several methods have emerged including Q-LoRA (Dettmers et al., 2023), CombLM (Ormazabal et al., 2023), and IPA (Lu et al., 2023). Concurrently, prompt-based learning methods have demonstrated effectiveness across various NLP tasks. Methods such as prompt-tuning (Lester et al., 2021), prefix-tuning (Li & Liang, 2021) and more recent approaches like proxy-tuning (Liu et al., 2024) and BBox-Adadpter (Sun et al., 2024) incorporate learnable continuous embeddings into the model's hidden states. They condition the frozen model to adapt to specific tasks without modifying the underlying architecture. Despite these advances, fine-tuning each updated LLM with PEFT to equip personalized knowledge remains highly costly, and how PEFT can bridge the gap in personalized settings within this evolving environment in a portable manner is yet to be fully explored. To this end, we develop in this paper the theory behind portable model patches that can be plugged into an evolved LLM to carry over the personalized knowledge from an earlier fine-tuned LLM.

## 3 PROPOSED METHOD

### 3.1 PRELIMINARIES

**Transformers.** Transformer models (Vaswani et al., 2023) is an architecture that has revolutionized NLP and other sequence-based tasks. It consists of two key components: (1) Multi-head self-attention and (2) feed-forward network. The multi-head self-attention mechanism is the core innovation of transformers. It computes the weighted representation of the input sequence where each token attends to every other token. Suppose we are given an input $X \in \mathbb{R}^{n \times d}$ where $n$ is the sequence length and $d$ is the hidden dimension of the transformer model. Then for any given attention head $i$ with a total of $H$ heads, we define the following three matrices: query matrix $W_{q_i} \in \mathbb{R}^{d \times d_H}$, key matrix $W_{k_i} \in \mathbb{R}^{d \times d_H}$, and value matrix $W_{v_i} \in \mathbb{R}^{d \times d_H}$, where $d_H = d/H$. Given these, we can

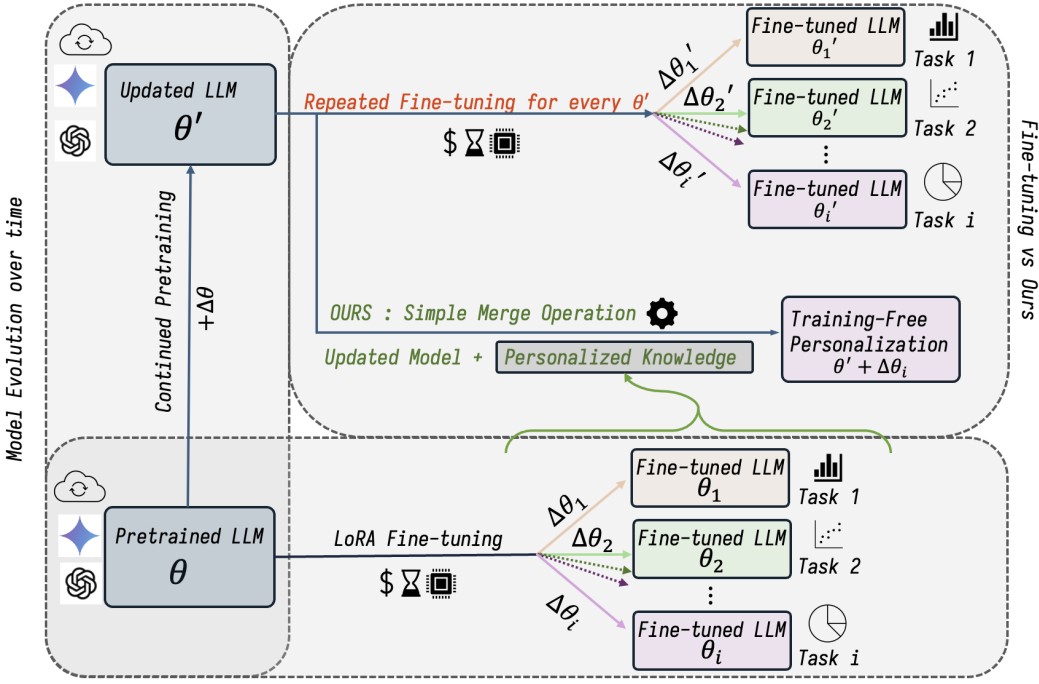

Figure 1: The diagram illustrates the core components of PORTLLM, a training-free framework to port personalized knowledge between evolving LLMs. Initially, a pretrained LLM is fine-tuned using LoRA. We transfer this task-specific knowledge without requiring the newer updated model to be fine-tuned again. This allows for continual performance improvements on downstream tasks without additional fine-tuning.

compute the self-attention for the $i$th head as follows:

$$h_i = \text{Softmax}\left(\frac{XW_{q_i}(XW_{k_i})^T}{\sqrt{d_H}}XW_{v_i}\right), \quad i = 1, \ldots, H. \tag{1}$$

The outputs of the $H$ attention heads are then concatenated as follows:

$$\text{Multi-head self-attention}(X) = \text{Concatenate}(h_1, h_2, \ldots, h_H)W_o, \tag{2}$$

where $W_o \in \mathbb{R}^{d \times d}$ is a projection matrix that combines the outputs of the different heads back into the model's hidden dimension $d$. After the multi-head self-attention, the output is passed through a position-wise feed-forward network that consists of two linear transformations and a nonlinear activation function (e.g., ReLU or GELU). Given the weight matrix of the first linear layer $W_{\text{up}} \in \mathbb{R}^{d \times d_m}$, weight matrix of the second layer $W_{\text{down}} \in \mathbb{R}^{d_m \times d}$, bias terms $b_1 \in \mathbb{R}^{d_m}$, $b_2 \in \mathbb{R}^d$, and a nonlinear activation function $\sigma(\cdot)$, where $d_m$ is the hidden dimension, the feed-forward network is applied independently to each position in the sequence as follows

$$\text{Feed-forward network}(X) = \sigma(XW_{\text{up}} + b_1)W_{\text{down}} + b_2. \tag{3}$$

Furthermore, these two layers are wrapped with residual connections and layer normalization.

**Low-Rank Adaptation (LoRA).** Consider a transformer model where $W_0 \in \mathbb{R}^{d \times d}$ is the pretrained weight matrix, which could be a weight matrix for any layer in the transformer. In a typical fine-tuning setup, the weights $W_0$ are updated during training to adapt the model to a specific task. This update requires storing and computing the entire matrix $W_0$ during training, which becomes computationally expensive for large models. Instead of updating the full weight matrix $W_0$, LoRA (Hu et al., 2021) assumes that the weight update $\Delta W \in \mathbb{R}^{d \times d}$, essentially the difference between pretrained weights $W_0$ and the hypothetical fine-tuned weight, can be approximated by a low-rank decomposition:

$$\Delta W = BA, \tag{4}$$

where $B \in \mathbb{R}^{d \times r}$ and $A \in \mathbb{R}^{r \times d}$ are trainable matrices, while $r$ is the rank of the decomposition with $r \ll d$. In this setup, the full weight update matrix $\Delta W$ is replaced by the product of two

smaller matrices, $B$ and $A$, drastically reducing the number of trainable parameters from $d^2$ to $2rd$. Hence, the fine-tuned model weights can be trained as follows:

$$W_{\text{new}} = W_0 + \Delta W = W_0 + BA. \tag{5}$$

By doing this, LoRA reduces the number of parameters that need to be trained while still allowing the model to adapt to new tasks.

## 3.2 PROPOSED TRAINING-FREE FRAMEWORK: PORTLLM

**Notations and Assumptions.** We refer to the pretrained LLM as the first version of the model, denoted by $\theta$, and the updated continued pretrained model as $\theta'$, as illustrated in Figure 2. We also assume that to get to $\theta'$, the provider does continued pretraining using LoRA, where $\Delta\theta$ denotes this adapter, however empirical experiments in Section 4.4 show that even if the provider does full weight continued pretraining on the newer dataset, our method still holds. We denote this full weight updated model as $\phi$. Similarly, for any downstream user $i$, we have a corresponding dataset denoted $d_i$. The base model fine-tuned on this dataset $d_i$ is denoted by $\theta_i$. We can also rewrite $\theta_i$ in terms of its LoRA update as $\theta_i = \theta + \Delta\theta_i$. Consequently, if we fine-tune updated model $\theta'$ for $i$th downstream task, we have $\theta_i' = \theta' + \Delta\theta_i'$. We further assume that a personalization adaptor $\Delta\theta_i$ or $\Delta\theta_i'$ has a rank that is much smaller than that of a continued pretrained adaptor $\Delta\theta$ or $\Delta\theta'$.

**Proposed Method of PORTLLM.** PORTLLM aims to approximate the fine-tuned updated model $\theta_i'$ by applying the older personalization adaptor/model patch $\Delta\theta_i$ to the continued pre-trained model $\theta'$. It will be shown in Section 3.3 that the older model patch $\Delta\theta_i$ may be used in lieu of the newer model patch $\Delta\theta_i'$, namely,

$$\theta_i' = \theta' + \Delta\theta_i' \approx \theta' + \Delta\theta_i. \tag{6}$$

In other words, one can readily add the extra knowledge $\Delta\theta_i$ from the previous fine-tuning process to the newest LLM $\theta'$. Throughout our experimental section, we perform experiments with this approximated patching process.

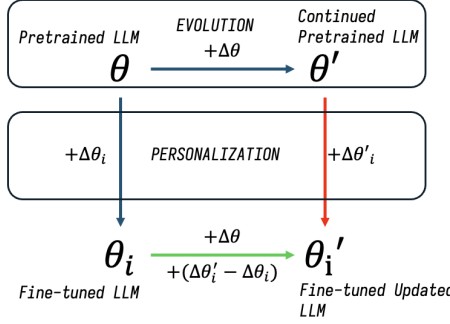

Figure 2: LLM's evolution & personalization cycle.

## 3.3 ANALYSIS OF OUR PROPOSED PORTABILITY

**Theoretical Justification.** The fine-tuned updated model $\theta_i'$ can be decomposed into a naive update term and a residual matrix $R$ as follows:

$$\theta_i' = \theta' + \Delta\theta_i' \tag{7a}$$

$$= \underbrace{(\theta' + \Delta\theta_i)}_{\text{Naive Update } \hat{\theta}_i'} + \underbrace{(\Delta\theta_i' - \Delta\theta_i)}_{\text{Residual Matrix } R}. \tag{7b}$$

*Lemma 1 (informal): We claim that the residual matrix $R$ is negligible compared to the naive update $\hat{\theta}_i'$. The key reason is that the model patches $\Delta\theta_i'$ and $\Delta\theta_i$ are both low rank (recall the assumption in Section 3.2), whereas the pretrained models $\theta$ and $\theta'$ are primarily full rank matrices. We provide a formal proof in Appendix C.*

**Empirical Validation.** We empirically show that the difference between two personalization updates, $R = \Delta\theta_i' - \Delta\theta_i$, is negligible when compared to the naive update term, $\theta' + \Delta\theta_i$. We use Frobenius norm, $\|\cdot\|_F$, and the maximum singular value, $\sigma_{\max}$, to measure the magnitudes of matrices. The results are summarized in Table 1 across four different downstream tasks {BoolQ, MRPC, RTE, WNLI}. We can see that the $\sigma_{\max}$ for first

Table 1: Comparison of the terms making up our framework across different datasets.

| Term | | BoolQ | MRPC | RTE | WNLI |
|---|---|---|---|---|---|
| $\theta' + \Delta\theta_i$ | $\sigma_{\max}$ | 7.37 | 7.37 | 7.37 | 7.37 |
| | $\|\cdot\|_F$ | 16.80 | 16.80 | 16.81 | 16.81 |
| $\Delta\theta_i' - \Delta\theta_i$ | $\sigma_{\max}$ | 0.19 | 0.14 | 0.10 | 0.08 |
| | $\|\cdot\|_F$ | 0.21 | 0.13 | 0.12 | 0.09 |
| $\sigma_{\max}/\sigma_{\max}$ | | 38.77 | 51.37 | 76.32 | 96.24 |
| $\|\cdot\|_F/\|\cdot\|_F$ | | 79.04 | 126.70 | 145.43 | 194.30 |

term are on average $66\times$ and the $\|\cdot\|_F$ is $136\times$ bigger in the favor of the first term, which implies that $R$ is comparatively negligible, hence our model patch can be simplified as shown in (6).

Table 2: Zero-shot performance comparison of model patches and baselines models on `Mistral-7B`, using the *OpenOrca* dataset for continued pretraining.

| Model Version | BoolQ Accuracy | SST-2 Accuracy | MRPC Accuracy/F1 | RTE Accuracy | WinoGrande Accuracy | WNLI Accuracy | GSM8K Accuracy |
|---|---|---|---|---|---|---|---|
| Pretrained LLM $\theta$ | 83.58 | 66.86 | 65.20 / 73.70 | 67.51 | 74.11 | 57.76 | 6.37 |
| Updated LLM $\theta'$ | 87.46 | 82.91 | 74.75 / 83.73 | 75.09 | 75.06 | 57.72 | 15.16 |
| Fine-tuned LLM $\theta_i$ | 91.01 | 95.99 | 89.46 / 92.62 | 87.73 | 85.95 | 83.11 | 34.04 |
| Fine-tuned Updated LLM $\theta'_i$ | 90.67 | 96.22 | 89.20 / 93.03 | 89.89 | 86.05 | 82.08 | 34.95 |
| $\theta' + \Delta\theta_i$ (Ours) | 90.24 | 96.10 | 88.73 / 92.10 | 89.17 | 85.01 | 83.10 | 41.32 |

## 4 EXPERIMENTS

**Datasets and Architecture.** We evaluate our framework on a diverse set of datasets to demonstrate PORTLLM's universality and effectiveness across various downstream tasks and domains. Specifically, we leverage datasets from the *GLUE* (Wang, 2018), *SuperGLUE* (Wang et al., 2019), *WinoGrande* (Sakaguchi et al., 2021) and *GSM8K* (Cobbe et al., 2021) benchmarks commonly used for such evaluation in the literature. For question answering tasks, we utilize *BoolQ* (from *SuperGLUE*) and *SST-2* (from GLUE); for similarity and paraphrase tasks, the *MRPC* (from *GLUE*) dataset; for inference tasks, the *RTE* and *WNLI* (both from *GLUE*) datasets; and lastly for reasoning tasks, we employ *WinoGrande* and *GSM8K*. This broad spectrum of tasks enables a comprehensive evaluation of our model's performance across diverse downstream applications. For continued pretraining datasets, we use the following: *OpenOrca* (Lian et al., 2023a), *SlimOrca* (Lian et al., 2023b), *OpenPlatypus* (Lee et al., 2023) and *AlpacaGPT4* (Peng et al., 2023). Additionally, we conduct extensive experiments across multiple model architectures to demonstrate the generalizability of our framework. Specifically, we test our method on `Mistral-7B` (Jiang et al., 2023), `Llama2-7B` (Touvron et al., 2023), `Llama3.1-8B` (Dubey et al., 2024), and `Gemma2-9B` (Team et al., 2024), showcasing the robustness and adaptability of our approach across different LLMs.

**Training Details.** To simulate the progression of time, we employ continued pretraining, where we transition from $\theta$ to $\theta'$ by taking a pretrained model and further pretraining it on a specific dataset using LoRA (Hu et al., 2021). In all continued pretraining scenarios, we maintain a constant rank $r = 64$ and $\alpha = 128$ with a learning rate of $0.0001$ and $4$ epochs. For downstream tasks, we also use LoRA, but in this case, we set the rank $r = 8$ consistently to ensure a fair comparison across tasks. Furthermore, for all LoRA applications, we optimize all the attention layers (Key, Value, Query, Projection) and all feed-forward network layers (Up Projection, Down Projection, and Gate Projection, where applicable). For each downstream fine-tuning, we use a constant learning rate of $0.0004$ while the number of epochs for each dataset {*BoolQ, SST-2, MRPC, RTE, WinoGrande, WNLI, GSM8K*} is {5, 5, 5, 5, 3, 5, 1}, respectively. Lastly, for fine-tuning on a specific downstream dataset mentioned, we solely use the train split and evaluate on the test split.

**Evaluation Metrics.** We use the *Language Model Evaluation Harness* (Gao et al., 2024a) by EleutherAI to assess the performance of all trained and fine-tuned models across the datasets in our experiments. All evaluations are conducted in a *zero-shot* setting rather than a few-shot setting. For datasets {*BoolQ, SST-2, RTE, WinoGrande, WNLI*}, we employ accuracy as the primary evaluation metric. In the case of {*MRPC*}, we utilize both accuracy and F1 score to provide a more comprehensive evaluation. For {*GSM8K*}, we had two evaluation options: (1) flexible match accuracy or (2) exact match accuracy. Due to the poor zero-shot performance of the models on *GSM8K* for exact matches, we opted for the flexible match accuracy.

### 4.1 SUPERIORITY OF PORTLLM FRAMEWORK

In this section, we compare the performance of our model patches against several baseline models, including the pretrained LLM $\theta$, the updated model using continued pretraining $\theta'$, the fine-tuned model $\theta_i$, and the updated fine-tuned model $\theta'_i$. For consistency, all models are variations of `Mistral-7B`, while the continued pretraining dataset is *OpenOrca*. Importantly, the performance

Table 3: Performance comparison of model patches $\Delta\theta_i$ across four downstream tasks {*BoolQ, MRPC, WNLI, WinoGrande*} using different continued pretraining datasets {*OpenOrca, SlimOrca, OpenPlatypus, AlpacaGPT4*}. All models are based on `Mistral-7B`.

| Dataset | Model | BoolQ Accuracy | MRPC Accuracy | WNLI Accuracy | WinoGrande Accuracy |
|---|---|---|---|---|---|
| OpenOrca | Updated Model $\theta'$ | 87.46 | 74.75 | 57.72 | 75.06 |
| | Ours $\theta' + \Delta\theta_i$ | 90.24 (↑ 2.78) | 88.73 (↑ 13.98) | 83.10 (↑ 25.38) | 85.01 (↑ 9.95) |
| SlimOrca | Updated Model $\theta'$ | 87.16 | 74.76 | 64.79 | 74.98 |
| | Ours $\theta' + \Delta\theta_i$ | 90.07 (↑ 2.91) | 87.75 (↑ 12.99) | 83.09 (↑ 18.30) | 85.59 (↑ 10.61) |
| OpenPlatypus | Updated Model $\theta'$ | 83.73 | 70.10 | 53.52 | 73.95 |
| | Ours $\theta' + \Delta\theta_i$ | 90.34 (↑ 6.61) | 90.20 (↑ 20.10) | 80.28 (↑ 26.76) | 83.58 (↑ 9.63) |
| AlpacaGPT4 | Updated Model $\theta'$ | 83.94 | 71.32 | 56.34 | 74.82 |
| | Ours $\theta' + \Delta\theta_i$ | 90.52 (↑ 6.58) | 89.22 (↑ 17.90) | 84.51 (↑ 28.17) | 84.93 (↑ 10.11) |

is evaluated under the zero-shot setting across all the datasets. The results are summarized in Table 2.

❶ Compared to the zero-shot accuracy of updated model $\theta'$, applying our model patches can result in significant performance gains, with improvements up to $2.7\times$. Notably, no additional training is required when applying these patches, as the process simply involves a merge operation. Across all evaluated datasets {BooLQ, SST-2, MRCP, RTE, WinoGrande, WNLI, GSM8K}, we observe substantial zero-shot performance gains of {2.7%, 13.19%, 13.98%, 14.08%, 9.95%, 25.38%, 26.16%}, respectively. This implies that our model patches are capable of transferring personalized knowledge across the different model versions.

❷ Comparing the performance of our model patches applied to the updated model $\theta' + \Delta\theta_i$ with that of the fine-tuned model $\theta_i$, we observe that our method successfully transfers most of the downstream task-specific knowledge, yielding comparable results. As shown in Table 2, for tasks {*BoolQ, MRPC, WinoGrande, WNLI*}, the performance is nearly identical, with a maximum difference of only $0.77\%$ in favor of the fine-tuned model. However, in tasks like {*SST-2, RTE, GSM8K*}, our approach outperforms the fine-tuned model by {0.11%, 1.44%, 7.28%}, respectively. These results suggest that, when paired with a pretraining dataset that enhances performance for a specific task, our model patches can further leverage this advantage to improve downstream task performance in certain cases.

❸ Moreover, our method performs on par with the fine-tuned updated model ($\theta' + \Delta\theta_i$ compared to $\theta'_i$), as evident from the comparison of the last two rows in Table 2. The difference between the two approaches is minimal when it comes to performance, with a maximum variation of just $1.04\%$ observed in the case of *WinoGrande*. Notably, while one method requires fine-tuning, our approach remains completely training-free. Additionally, for certain downstream tasks such as *WNLI* and *GSM8K*, our method outperforms the fine-tuned updated model by $1.02\%$ and $6.37\%$, respectively. This demonstrates that our approach not only provides comparable results but can, in some instances, surpass the performance of a fine-tuned evolved model.

## 4.2 CONSISTENT RESULTS ACROSS DIFFERENT PRETRAINING DATASETS

In this section, we investigate the impact of different pretraining datasets on our model patches $\Delta\theta_i$ and assess whether our method can effectively leverage updates obtained through continued pretraining. We conduct a comparative analysis across four downstream datasets – {*BoolQ, MRPC, WNLI, WinoGrande*} – alongside four distinct continued pretraining datasets: *OpenOrca*, *SlimOrca*, *OpenPLatypus*, and *AlpacaGPT4*. Consistent with our previous section, all experiments utilize the `Mistral-7B` model. The results of this analysis are summarized in Table 3.

❶ The results presented in Table 3 demonstrate that our model patches exhibit strong portability across different downstream tasks, irrespective of the continued pretraining dataset used. For each specific downstream task, we observe substantial improvements in zero-shot performance compared to the updated model $\theta'$ across all pretraining datasets. For instance, in the case of *WNLI*, the perfor-

Table 4: Performance analysis of model patches across various architectures {Mistral-7B, Llama2-7B, Llama3.1-8B, Gemma2-9B} on four downstream tasks {*BoolQ, MRPC, WNLI, WinoGrande*} under four different model settings. For each downstream task $\boxed{\text{Cyan}}$ highlights the best performance for each model architecture.

| Model | Version | BoolQ Accuracy | MRPC Accuracy/F1 | WNLI Accuracy | WinoGrande Accuracy |
|---|---|---|---|---|---|
| Mistral 7B | Pretrained Model $\theta$ | 83.58 | 65.20 / 73.70 | 57.76 | 74.11 |
| | Updated Model $\theta'$ | 87.46 | 74.75 / 83.73 | 57.72 | 75.06 |
| | Fine-tuned Model $\theta_i$ | 91.01 | 89.46 / 92.62 | 83.11 | 85.95 |
| | Ours $\theta' + \Delta\theta_i$ | 90.24 | 88.73 / 92.10 | 83.10 | 85.01 |
| Llama 2 7B | Pretrained Model $\theta$ | 77.74 | 69.12 / 81.52 | 45.07 | 69.06 |
| | Updated Model $\theta'$ | 82.69 | 69.61 / 81.60 | 47.89 | 70.48 |
| | Fine-tuned Model $\theta_i$ | 88.21 | 88.97 / 92.01 | 57.75 | 75.45 |
| | Ours $\theta' + \Delta\theta_i$ | 88.32 | 89.95 / 92.56 | 53.77 | 76.87 |
| Llama 3.1 8B | Pretrained Model $\theta$ | 82.35 | 66.91 / 77.54 | 59.15 | 73.56 |
| | Updated Model $\theta'$ | 85.63 | 75.00 / 83.60 | 61.95 | 73.64 |
| | Fine-tuned Model $\theta_i$ | 90.22 | 84.31 / 89.51 | 83.10 | 85.71 |
| | Ours $\theta' + \Delta\theta_i$ | 90.03 | 89.71 / 92.71 | 81.69 | 84.85 |
| Gemma 2 9B | Pretrained Model $\theta$ | 83.98 | 68.63 / 74.19 | 57.75 | 74.19 |
| | Updated Model $\theta'$ | 88.41 | 77.21 / 84.93 | 74.65 | 76.72 |
| | Fine-tuned Model $\theta_i$ | 91.38 | 91.42 / 93.83 | 83.20 | 83.98 |
| | Ours $\theta' + \Delta\theta_i$ | 91.16 | 90.69 / 93.17 | 88.73 | 83.82 |

mance boosts are {25.38%, 18.30%, 26.76%, 28.17%} for {*OpenOrca, SlimOrca, OpenPlatypus, AlpacaGPT4*} datasets, respectively. A similar trend of significant improvement is also evident across other downstream tasks.

❷ We further observe from Table 3 that certain pretraining datasets can either enhance or detract from the zero-shot performance of $\theta'$ on specific downstream tasks, and this effect carries over to our frameworks to some degree. For instance, continued pretraining on *OpenPlatypus* leads to a decrease in performance on the *WNLI* dataset. Consequently, the addition of our model patch in this scenario results in the lowest accuracy for this particular downstream task among all the pretraining datasets evaluated.

## 4.3 CONSISTENT RESULTS ACROSS DIFFERENT MODEL ARCHITECTURES

This section provides a comprehensive analysis of our model patches across various architectures to evaluate their performance. We examine four different model architectures: {Mistral-7B, Llama2-7B, Llama3.1-8B, Gemma2-9B}, assessing their effectiveness on four distinct downstream tasks {*BoolQ, MPRC, WNLI, WinoGrande*}. Performance is evaluated under four settings: (1) pretrained model, (2) continued pretrained model or updated model, (3) fine-tuned model, and (4) our model patches ported to the updated model. The *OpenOrca* dataset is used for our continued pretraining in this analysis. The results are summarized in Table 4.

❶ The results across different model architectures indicate that our training-free model patches significantly enhance zero-shot performance for all downstream tasks. In each case, our approach matches personalized performance (fine-tuned model), and in certain instances, it even surpasses it. For example, with Gemma2-9B, when the personalized performance exceeds our method, the difference in accuracy is at most $0.73\%$, which can be considered negligible. Conversely, in scenarios where our method outperforms personalized performance, we observe improvements of up to $5.53\%$. A similar trend is noted across the other model architectures, as detailed in Table 4.

❷ Additionally, we observe that fine-tuning is essential for achieving optimal zero-shot performance on downstream tasks. Across all model architectures, the zero-shot performance of both the pre-

Table 5: Evaluation of model patches added to `Mistral-7B` with full weight continued pretraining on the OpenOrca dataset across various downstream tasks.

| Model Version | BoolQ Accuracy | SST-2 Accuracy | MRPC Accuracy/F1 | RTE Accuracy | WinoGrande Accuracy | WNLI Accuracy | GSM8K Accuracy |
|---|---|---|---|---|---|---|---|
| Full Weight Updated Model $\phi$ | 86.61 | 93.81 | 77.21 / 85.31 | 75.09 | 72.77 | 63.38 | 20.55 |
| $\phi + \Delta\theta_i$ (Ours) | 89.88 | 95.53 | 87.25 / 90.97 | 90.61 | 85.08 | 80.28 | 41.24 |

trained model $\theta$ and the updated model $\theta'$ is subpar, with particularly poor results noted on tasks like *WNLI*. This reinforces the notion that excellent performance necessitates some form of fine-tuning, further motivating the need for our training-free framework. Another significant finding is the slight performance improvement for downstream tasks across all model architectures due to continued pretraining. Therefore, it is advisable for the downstream user to utilize the updated model weights, as this may provide beneficial enhancements in performance.

## 4.4 PORTLLM ALSO WORKS WITH FULL WEIGHT CONTINUED PRETRAINING

For our theoretical analysis, we initially assumed that continued pretraining was conducted using LoRA. However, we aim to investigate whether our method is effective across model evolution when the updates occur through full weight continued pretraining. Such a model is denoted $\phi$. To evaluate this, we utilize `Mistral-7B`, which has undergone full weight continued pretraining on the OpenOrca dataset, and incorporate our model patches for various downstream tasks. We then compare the performance of these patched models against the zero-shot performance of $\phi$ to assess the improvements attributable to our model patches. The results across various downstream tasks are summarized in Table 5.

We find that our model patches can be effectively applied to a continued pretrained model utilizing full weight updates rather than relying solely on LoRA. Across all evaluated datasets – {*BoolQ, SST-2, MRPC, RTE, WinoGrande, WNLI, GSM8K*} – we observe significant performance improvements of {$3.27\%, 1.72\%, 10.04\%, 15.52\%, 12.31\%, 16.90\%, 20.69\%$}, respectively, compared to the zero-shot performance of the updated model.

## 4.5 COMPUTING EFFICIENCY COMPARISON OF PORTLLM

This subsection evaluates the performance of our method from an efficiency perspective. We employ the following metrics for comparison: (1) Number of trainable parameters, (2) GPU memory utilization,

Table 6: Efficiency comparison between PORTLLM and LoRA on *SST-2* with `Mistral-7B` as the model architecture. The table compares trainable parameters, GPU Memory Usage, and GPU Hours for PORTLLM and LoRA fine-tuning.

| Metric | Ours $\theta' + \Delta\theta_i$ | Fine-tuning Model $\theta_i$ | Savings |
|---|---|---|---|
| # Trainable Parameters | 0 | $20,971,520$ | 100% |
| GPU Memory Utilization (GB) | 28.71 | 350.61 | 12.21$\times$ |
| GPU Hours | 0.0083 | 40.65 | 4897$\times$ |

and (3) GPU hours. We analyze the merging of our model patches in relation to model fine-tuning using LoRA to achieve comparable performance. For LoRA fine-tuning calculations, we have the following settings: Downstream task *SST-2* for `Mistral-7B`, with local batch size of 4 and 5 epochs. The results are summarized in Table 6.

Compared to downstream fine-tuning using LoRA, our method offers a plug-and-play solution with no trainable parameters, resulting in a reduction of nearly 20 million in parameters that need to be trained. This training-free paradigm not only conserves resources but also saves up to 12.2$\times$ GPU memory, reducing the requirement from 350 GB for LoRA to just 28.7 GB during the merge operation of model patches. Additionally, the merge operation can be executed in mere seconds, in contrast to the hours required for fine-tuning. This opens many other doors for applications of our model patches. PORTLLM demonstrates the potential for on-device, training-free models for various downstream tasks without the need for fine-tuning. Furthermore, it reduces the need for expensive cloud infrastructure, especially in large-scale fine-tuning.

## 5    CONCLUSION

In this paper, we propose PORTLLM, a framework aimed at addressing the challenges faced by downstream users of pretrained LLMs when adapting to frequent model evolutions over time. By leveraging lightweight model patches, PORTLLM offers a training-free, cost-effective solution to seamlessly transfer domain-specific knowledge between different iterations of LLMs. This enables users to maintain, and sometimes even enhance, their models' performance on specialized tasks without the need for repeated fine-tuning or extensive computational resources. Through extensive empirical evaluations across a set of tasks and models, we demonstrate that our method not only preserves performance but can also leverage the continual updates in pretrained LLMs, offering substantial gains in task-specific performance. Moreover, we provide theoretical insights into the portability of these model patches, highlighting the underlying factors that make them effective across evolving model versions. Looking forward, PORTLLM paves the way for more robust and adaptable solutions in the evolving landscape of LLM personalization, offering another avenue for training-free adaptation. Furthermore, future endeavours will aim at developing such methods that work across different model architecture, including using techniques from model merging.

## 6    REPRODUCIBILITY STATEMENT

To ensure reproducibility, we provide detailed descriptions of datasets, model architectures, training settings, and evaluation metrics used in our experiments in Section 4. The same section also dives deep into the hyperparameters used for all the tasks mentioned in our paper, including LoRA fine-tuning and downstream evaluation. Furthermore, we have also provided all the training scripts alongside the hyperparameters as supplementary material so that results from our papers can be reproduced with minimal effort. Lastly, the datasets and model architectures utilized in this paper are open-source and publicly available for anyone's use. Each dataset, as well as model, have been cited accordingly so that anyone can reproduce the experiments.

## ACKNOWLEDGMENT

This research was, in part, funded by the CISCO Faculty Award, UNC SDS Seed Grant and Net-Mind.AI. The views and conclusions contained in this document are those of the authors and should not be interpreted as representing official policies, either expressed or implied of the funding organizations.

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

APPENDIX

# A ANALYSIS OF MODEL PERFORMANCE UNDER MULTIPLE CONTINUAL UPDATES

To validate our framework's robustness under periodic updates, we conducted experiments simulating multiple rounds of continued pretraining. Using `Mistral-7B` as our base model, we performed four sequential updates using different pretraining datasets: OpenOrca → OpenPlatypus → Alpaca → GPT4-LLM-Cleaned. For the hyperparameters, we utilize the same settings described in Section 4. We evaluated our model patch on all seven downstream tasks after each update. The results are summarized in Table A. Our experiments yield the following key findings:

Table A: Model performance across sequential updates ($T = 0$ to $T = 4$) on seven downstream tasks. We report accuracy for all tasks except MRPC, where we show both accuracy and F1 score.

| Time | Model Version | BoolQ Accuracy | SST-2 Accuracy | MRPC Accuracy/F1 | RTE Accuracy | WinoGrande Accuracy | WNLI Accuracy | GSM8K Accuracy |
|---|---|---|---|---|---|---|---|---|
| $T = 0$ | Pretrained Model $\theta$ | 83.58 | 66.86 | 65.20 / 73.70 | 67.51 | 74.11 | 57.76 | 6.37 |
| None | Fine-tuned Model $\theta_i$ | 91.01 | 95.99 | 89.46 / 92.62 | 87.73 | 85.95 | 83.11 | 34.04 |
| $T = 1$ | Updated Model $\theta'$ | 87.46 | 82.91 | 74.75 / 83.73 | 75.09 | 75.06 | 57.72 | 15.16 |
| OpenOrca | Ours $\theta' + \Delta\theta_i$ | 90.24 | 96.11 | 88.73 / 92.10 | 89.17 | 85.01 | 83.10 | 41.32 |
| $T = 2$ | Updated Model $\theta'$ | 86.33 | 86.81 | 76.23 / 83.53 | 71.48 | 74.51 | 52.11 | 12.36 |
| OpenPlatypus | Ours $\theta' + \Delta\theta_i$ | 89.88 | 96.22 | 88.24 / 91.55 | 88.09 | 84.37 | 83.10 | 42.15 |
| $T = 3$: | Updated Model $\theta'$ | 87.03 | 85.78 | 74.02 / 83.33 | 73.65 | 74.27 | 56.34 | 16.38 |
| Alpaca | Ours $\theta' + \Delta\theta_i$ | 89.66 | 96.33 | 88.73 / 92.12 | 88.81 | 85.08 | 83.10 | 38.21 |
| $T = 4$: | Updated Model $\theta'$ | 85.11 | 75.79 | 72.78 / 82.79 | 74.37 | 71.67 | 56.34 | 14.94 |
| GPT4-LLM | Ours $\theta' + \Delta\theta_i$ | 88.41 | 96.10 | 87.01 / 90.91 | 85.56 | 80.19 | 77.46 | 31.77 |

❶ Across all update stages ($T = 0$ to $T = 4$), applying our model patches results in substantial zero-shot improvements over the updated model $\theta'$. These improvements are consistent and significant across different tasks, with SST-2 showing gains from $+9.41\%$ to $+20.31\%$, WNLI maintaining strong improvements between $+21.12\%$ and $+30.99\%$, and MRPC consistently improving by $+12$ to $14\%$ in accuracy. Notably, these improvements are achieved without any additional training, requiring only a simple merge operation of our model patches.

❷ The performance stability of our patched models is particularly noteworthy when compared to the fluctuating zero-shot performance of $\theta'$. As shown in Table A, our method maintains remarkably consistent performance across multiple updates. For instance, BoolQ accuracy remains within a tight range of $88.41$ to $90.24\%$, SST-2 consistently maintains accuracy above $96\%$, and MRPC's F1 score stays above 90 across all update stages. These results demonstrate our method's robustness to successive model updates and its ability to preserve task-specific knowledge.

❸ Furthermore, our method shows interesting behavior in leveraging complementary knowledge from different updates. Taking GSM8K as an example, we observe varying but significant improvements ranging from $+16.83\%$ to $+29.79\%$ across different update stages. This suggests that our model patches can effectively combine knowledge from both the original fine-tuning and the continued pretraining updates, sometimes leading to performance gains that exceed what might be expected from either source alone. Such behavior demonstrates the potential of our approach to not just preserve but potentially enhance task performance through knowledge integration across model versions.

# B ANALYSIS OF LoRA RANK SELECTION FOR DOWNSTREAM TASKS

To validate our choice of LoRA rank and understand its impact on model performance, we conducted experiments with varying ranks. This analysis helps establish the optimal balance between computational efficiency and model effectiveness. To perform these experiments, we utilize `Mistral-7B` with OpenOrca as the continued pretraining dataset and evaluate on BoolQ, MRPC, and WNLI downstream tasks with a varying rank for LoRA. The rest of the hyperparameters are the same as in Section 4. The results are shown in Table B.

❶ Both fine-tuning $\theta_i$ and our method $\theta' + \Delta\theta_i$ demonstrate remarkable stability across different ranks, with optimal performance typically achieved at rank 8 to 16. Specifically, on BoolQ, we observe peak accuracy at rank 16, while MRPC and WNLI show optimal performance at rank 8. This consistency across ranks validates the robustness of our approach and suggests that model patches effectively capture task-specific knowledge regardless of rank selection.

❷ When examining the efficiency aspects, we find that increasing rank beyond 8 provides diminishing or even negative returns. As shown in Table B, at rank 32, we observe decreased performance across most tasks compared to rank 8: BoolQ drops by $1.01\%$, MRPC by $0.49\%$, and WNLI by $2.83\%$. Importantly, the performance gap between our method and direct fine-tuning remains minimal even at lower ranks, with differences of less than $1\%$ in most cases. This suggests that our training-free approach maintains its effectiveness even with more constrained rank settings.

❸ Our analysis strongly justifies our original choice of rank 8 as the default setting. This configuration achieves an optimal balance between computational efficiency (requiring fewer parameters than higher ranks), model performance (maintaining competitive results across all tasks), and adaptation capability (providing sufficient capacity for task-specific learning). Notably, while higher ranks like 16 or 32 require significantly more parameters, they offer minimal or no performance benefits, making rank 8 the sweet spot for our training-free framework.

Table B: Performance comparison across different LoRA ranks (2, 4, 8, 16, 32) on three downstream tasks using `Mistral-7B`.

| Rank | Model Version | BoolQ Accuracy | MRPC Accuracy/F1 | WNLI Accuracy |
|---|---|---|---|---|
| $r = 2$ | Fine-tuned Model $\theta_i$ | 90.52 | 87.30 / 91.28 | 74.65 |
|  | Ours $\theta' + \Delta\theta_i$ | 89.85 | 87.01 / 91.03 | 77.46 |
| $r = 4$ | Fine-tuned Model $\theta_i$ | 90.81 | 89.42 / 91.98 | 78.23 |
|  | Ours $\theta' + \Delta\theta_i$ | 90.18 | 87.75 / 91.53 | 80.28 |
| $r = 8$ | Fine-tuned Model $\theta_i$ | 91.01 | 89.46 / 92.62 | 83.11 |
|  | Ours $\theta' + \Delta\theta_i$ | 90.24 | 88.73 / 92.10 | 83.10 |
| $r = 16$ | Fine-tuned Model $\theta_i$ | 91.07 | 89.22 / 92.49 | 81.69 |
|  | Ours $\theta' + \Delta\theta_i$ | 90.98 | 88.97 / 92.31 | 82.98 |
| $r = 32$ | Fine-tuned Model $\theta_i$ | 90.00 | 88.97 / 92.15 | 80.28 |
|  | Ours $\theta' + \Delta\theta_i$ | 90.28 | 88.73 / 91.93 | 81.69 |

## C  PROOF OF LEMMA 1

Notations: $\mathcal{C}(\cdot)$ returns the column vector subspace of a matrix.

Recall in Section 3.3, we decomposed the fine-tuned updated model $\theta_i'$ into two terms:

$$\theta_i' = \theta' + \Delta\theta_i' \tag{8a}$$
$$= \underbrace{(\theta' + \Delta\theta_i)}_{\text{Naive Update } \hat{\theta}_i'} + \underbrace{(\Delta\theta_i' - \Delta\theta_i)}_{\text{Residual Matrix } R}. \tag{8b}$$

*Lemma 1: The residual matrix $R$ is negligible compared to the naive update $\hat{\theta}_i'$ in terms of the Frobenius norm.*

*Proof:* Our goal is to show that the error ratio $\|R\|_{\text{F}}^2 / \left\|\hat{\theta}_i'\right\|_{\text{F}}^2$ is small. We will proceed by finding a large enough numerator $\|R\|_{\text{F}}^2$ and small enough $\left\|\hat{\theta}_i'\right\|_{\text{F}}^2$ in the LoRA context and show that the upper bound of the error ratio is small.

**Numerator** $\|R\|_{\text{F}}^2$. First, we search for conditions that potentially lead to residual matrices with larger Frobenius norm. We apply compact SVD to the patches $\Delta\theta_i$ and $\Delta\theta_i'$, designating subscripts

1 and 2 for SVD matrices, respectively:

$$\|R\|_{\mathrm{F}}^2 = \|\Delta\theta_i' - \Delta\theta_i\|_{\mathrm{F}}^2 \tag{9a}$$

$$= \left\|U_2\Sigma_2 V_2^T - U_1\Sigma_1 V_1^T\right\|_{\mathrm{F}}^2 \tag{9b}$$

$$= \left\|\sum_{\ell=1}^{r_2} \sigma_\ell^{(2)} u_\ell^{(2)} v_\ell^{(2)T} - \sum_{k=1}^{r_1} \sigma_k^{(1)} u_k^{(1)} v_k^{(1)T}\right\|_{\mathrm{F}}^2. \tag{9c}$$

Here, the typical order of magnitude for $r_1$ and $r_2$ is about 10.

1. When there is no intersection between the two pairs of singular vector subspaces, namely, $\mathcal{C}(U_1)\cap \mathcal{C}(U_2) = \varnothing$ and $\mathcal{C}(V_1)\cap\mathcal{C}(V_2) = \varnothing$, the two terms in (9c) may be combined to form a valid compact SVD of rank $r_1 + r_2$ as follows:

$$\|R\|_{\mathrm{F}}^2 = \left\|\sum_{\ell=1}^{r_2} \sigma_\ell^{(2)} u_\ell^{(2)} v_\ell^{(2)T} + \sum_{k=1}^{r_1} \sigma_k^{(1)} \cdot (-u_k^{(1)}) v_k^{(1)T}\right\|_{\mathrm{F}}^2 \tag{10a}$$

$$= \left\|\sum_{k'=1}^{r_1+r_2} \sigma_{k'}^{(1,2)} u_{k'} v_{k'}^T\right\|_{\mathrm{F}}^2, \tag{10b}$$

where $\sigma_1^{(1,2)},\ldots,\sigma_{r_1+r_2}^{(1,2)}$ is a list of descending ordered positive numbers sampled without replacement from $\{\sigma_k^{(1)}\}_{k=1}^{r_1}$ and $\{\sigma_\ell^{(2)}\}_{\ell=1}^{r_2}$. Applying the Frobenius norm property to the SVD representation of a matrix, we obtain

$$\|R\|_{\mathrm{F}}^2 = \sum_{k'=1}^{r_1+r_2} \left[\sigma_{k'}^{(1,2)}\right]^2 = \sum_{\ell=1}^{r_2} \left[\sigma_\ell^{(2)}\right]^2 + \sum_{k=1}^{r_1} \left[\sigma_k^{(1)}\right]^2. \tag{11}$$

This is the case when the two patches $\Delta\theta_i$ and $\Delta\theta_i'$ contain only orthogonal information. This is not very realistic because the two patches were created on the same downstream task $i$ that should lead to some information in common.

2. When the two patches have are oppositely embedded in one of the singular value subspaces, e.g., $U_2 = -U_1$ and $V_2 = V_1$, the two terms in (9c) can be merged and singular values with the same ranking will be summed up, namely,

$$\|R\|_{\mathrm{F}}^2 = \left\|\sum_{\ell=1}^{r_2} \sigma_\ell^{(2)} u_\ell^{(2)} v_\ell^{(2)} + \sum_{k=1}^{r_1} \sigma_k^{(1)} \cdot (-u_k^{(1)}) v_k^{(1)T}\right\|_{\mathrm{F}}^2 \tag{12a}$$

$$= \left\|\sum_{\ell=1}^{r_2} \left[\sigma_\ell^{(2)} + \sigma_\ell^{(1)}\right] u_\ell^{(2)} v_\ell^{(2)}\right\|_{\mathrm{F}}^2 \tag{12b}$$

$$= \sum_{\ell=1}^{r_2} \left[\sigma_\ell^{(2)} + \sigma_\ell^{(1)}\right]^2, \tag{12c}$$

which can be easily shown that it is larger than the orthogonal case (11) due to the extra interaction term $\sum_{\ell=1}^{r_2} \sigma_\ell^{(2)}\sigma_\ell^{(1)}$. When $\sigma_\ell^{(2)} = \sigma_\ell^{(1)}$, this case corresponds to two patches having exactly opposite update directions, which again, is not very realistic because the same downstream tasks are used to generate the update directions. Equations (11) and (12c) both correspond to extreme conditions, and the continuum in between should be more realistic. We will use the large numerator (12c) to examine the error ratio.

**Denominator** $\left\|\hat{\theta}'_i\right\|_F^2$. We continue to apply compact SVD to the continued pretrained model $\theta'$ and patch $\Delta\theta_i$, designating subscripts 0 and 1 for SVD matrices, respectively:

$$\left\|\hat{\theta}'_i\right\|_F^2 = \|\theta' + \Delta\theta_i\|_F^2 \tag{13a}$$

$$= \left\|U_0\Sigma_0 V_0^T + U_1\Sigma_1 V_1^T\right\|_F^2 \tag{13b}$$

$$= \left\|\sum_{j=1}^{r_0} \sigma_j^{(0)} u_j^{(0)} v_j^{(0)T} + \sum_{k=1}^{r_1} \sigma_k^{(1)} u_k^{(1)} v_k^{(1)T}\right\|_F^2. \tag{13c}$$

Here, the typical order of magnitude for $r_0$ is about 100.

1. When there is no intersection between the two pairs of singular vector subspaces, the two terms may be combined to form a valid compact SVD of rank $r_0 + r_1$. Hence, similar to (11), we have

$$\left\|\hat{\theta}'_i\right\|_F^2 = \sum_{j=1}^{r_0} \left[\sigma_j^{(0)}\right]^2 + \sum_{k=1}^{r_1} \left[\sigma_k^{(1)}\right]^2. \tag{14}$$

2. When the basis vectors of singular matrices (with one matrix having opposite signs) of the patch can be found in the singular matrices of the continued pretrained model, we are able to combine the two terms in (13c) by using the basis vectors of $U_0$ and $V_0$ as follows:

$$\left\|\hat{\theta}'_i\right\|_F^2 = \left\|\sum_{j=1}^{r_0} \sigma_j^{(0)} u_j^{(0)} v_j^{(0)T} + \sum_{j=1}^{r_0} \sigma_j^{(1')} u_j^{(0)} v_j^{(0)T}\right\|_F^2, \tag{15a}$$

$$= \sum_{j=1}^{r_0} \left[\sigma_j^{(0)} - \sigma_j^{(1')}\right]^2, \tag{15b}$$

where we define an auxiliary symbol

$$\sigma_j^{(1')} = \begin{cases} \sigma_k^{(1)}, & \exists k \in [1, r_1] \text{ s.t. } u_j^{(0)} = u_k^{(1)}, \\ 0, & \text{other } k. \end{cases} \tag{16}$$

Both (14) and (15b) correspond to the extreme cases. Since (15b) leads to a smaller denominator, we will use it to examine the error ratio.

**Error Ratio.** Using (12c) and (15b), a pessimistic error ratio can be approximated and then up bounded as follows:

$$\frac{\|R\|_F^2}{\left\|\hat{\theta}'_i\right\|_F^2} \approx \frac{\sum_{\ell=1}^{r_2} \left[\sigma_\ell^{(2)} + \sigma_\ell^{(1)}\right]^2}{\sum_{j=1}^{r_0} \left[\sigma_j^{(0)} - \sigma_j^{(1')}\right]^2} \tag{17a}$$

$$\leq \frac{r_2 \cdot \max_\ell \left[\sigma_\ell^{(2)} + \sigma_\ell^{(1)}\right]^2}{r_0 \cdot \min_j \left[\sigma_j^{(0)} - \sigma_j^{(1')}\right]^2} \tag{17b}$$

$$= \frac{r_2}{r_0} \cdot \nu, \tag{17c}$$

where $\nu = \max_\ell \left[\sigma_\ell^{(2)} + \sigma_\ell^{(1)}\right]^2 \Big/ \min_j \left[\sigma_j^{(0)} - \sigma_j^{(1')}\right]^2$ is a singular-value based constant. Given that the rank $r_2$ of the patch is at least one order of magnitude smaller than the rank $r_0$ of the pretrained model, we conclude the residual matrix term is negligible compared to the naive update term.

