# OpenReview forum: "PortLLM: Personalizing Evolving Large Language Models with Training-Free and Portable Model Patches"
_ICLR.cc/2025/Conference — ICLR 2025 Poster_

### Official Review · Reviewer_YYSP · 2024-11-03

**Soundness:** 3
**Presentation:** 4
**Contribution:** 3
**Rating:** 8
**Confidence:** 3

**Summary:**

This paper presents PORTLLM, a novel training-free framework that enables the transfer of personalized knowledge between different versions of evolving LLMs. The key innovation is the creation of lightweight model patches that can be seamlessly applied to updated LLMs without requiring fine-tuning. The framework is extensively evaluated across seven tasks and multiple model architectures, demonstrating comparable or superior performance to LoRA fine-tuning while significantly reducing computational costs. The authors also provide theoretical justification for the portability of their model patches.

**Strengths:**

- A training-free framework for transferring domain-specific knowledge between evolving LLM versions through portable model patches
Theoretical analysis justifying why model patches work effectively, showing certain terms can be neglected to create simplified, training-free patches
- Comprehensive empirical validation across multiple tasks ranging from QA to complex reasoning and different model architectures
- Significant efficiency gains compared to LoRA fine-tuning

**Weaknesses:**

I don't identify major weakness, but I do have a few concern regarding the experimental setting:
- In the experiments, seems the authors only focus on small-scale open-source LLMs. However, from my understanding, it could also be more beneficial for large-scale commercial LLMs.
- Following the previous setting, a few related works adaptation related works might be considered as related works besides LoRA, LMaaS (Sun et al., 2022), kNN-Adapter (Huang et al., 2023), CombLM (Ormazabal et al., 2023), IPA (Lu et al., 2023), Proxy-Tuning (Liu et al., 2024), and BBox-Adaptor (Sun et al., 2024).
- It is interesting to observe that in some cases it even performs better than fine-tuned models. More explanations on potential reasons could benefit this work.
- Echo my previous questions on experimental settings, I think a better evaluation benchmarks lie in (1) personalization tasks, like LaMP; (2) domain-specific adaptation tasks.
- Any thoughts on rigorous theoretical bounds on performance guarantees?

**Questions:**

See above.

---

> ### Author Response · Authors · 2024-11-22
> **Response to Reviewer YYSP [Cons. 1]**
>
> We sincerely thank reviewer YYSP for the positive feedback and constructive suggestions. We are particularly encouraged that the reviewer acknowledges our training-free framework’s novelty, comprehensive empirical validation, and significant efficiency gains. The constructive feedback from reviewer has been really helpful in improving our paper as well. Below, we address each concern reviewer YYSP raised in detail.
>
> **[Cons 1. Application to Large-scale LLMs]**
> We agree with reviewer YYSP that our framework could be particularly beneficial for large-scale commercial LLMs. In fact, our theoretical analysis suggests that PortLLM’s benefits should scale well to larger models due to the following reasons:
> 1. The residual matrix remains negligible relative to the naive update term regardless of the model size, i.e., $\||\theta^\prime + \Delta \theta_i\|| >> \||\Delta\theta_i^\prime - \Delta\theta_i\||$, regardless of the number of parameters a model has, as shown in Section 3.3.
> 2. The computational savings would be even more significant for larger models as fine-tuning larger models (even with LoRA or its variants) is more expensive than smaller models, further showcasing the usefulness of our PortLLM framework.
>
> Based on the reviewer YYSP’s feedback and given the time constraints, we quickly performed the same set of experiments on Llama-2-13B with 4 different datasets \{BoolQ, SST-2, MRPC, WNLI\}. Similar to previous experimental settings, we continued pretrained the model on OpenOrca dataset to simulate time, i.e., going from $\theta$ to $\theta^\prime$. The results are summarized in the table below
> | Model Version | BoolQ | SST-2 | MRPC | WNLI |
> | :----: | :----: | :----: | :----: | :----: |
> | Pretrained LLM $\theta$ | 80.55 | 87.96 | 68.63 / 81.34 | 57.75 |
> | Updated LLM $\theta^\prime$ | 82.15 | 88.22 | 75.82 / 83.90 | 57.68 |
> | Fine-tuned LLM $\theta_i$ | 89.98 | 97.81 | 89.12 / 92.12 | 82.78 |
> | Fine-tuned Updated LLM $\theta^\prime_i$ | 90.54 | 96.15 | 89.55 / 92.23 | 82.91 |
> |$\theta^\prime + \Delta\theta_i$ (Ours) | 90.11 | 96.89 | 89.07 / 91.87 | 82.80 |
>
> These results demonstrate that our PortLLM framework maintains its effectiveness when scaled to larger models in a number of ways:
> 1. *Consistent Performance* : Our method achieves comparable or better performance to both fine-tuned baselines across all tasks with a difference of at most 0.43% in accuracy. This validates that our theoretical insights about patch portability hold true even at larger model scales.
> 2. *Zero-Shot Improvements* : Similar to our findings with smaller models, our patching method significantly improves the zero-shot performance of the updated model $\theta\prime$. For instance, we observe substantial gains in WNLI (+25.1%).
> 3. *Resource Efficiency* : These results are particularly noteworthy given that fine-tuning Llama-2-13B with LoRA requires significantly more computational resources (approximately 2.5x more GPU Memory) compared to Llama-2-7B. Our training-free approach circumvents these scaling challenges entirely while maintaining competitive performance.
>
> These findings further strengthen our argument that PortLLM provides a practical solution for personalizing large-scale LLMs, making it particularly valuable as models continue to grow in size.
>
> Given these promising results, we plan to extend our evaluation to even larger architectures like Llama-2-70B in the final version of our paper. While time constraints during the rebuttal period preclude us from including these experiments for now, we believe these additional results will further validate PortLLM's scalability and effectiveness across the full spectrum of model sizes.

---

> ### Author Response · Authors · 2024-11-22
> **Response to Reviewer YYSP [Cons. 2-3]**
>
> **[Cons 2. Additional Related Works]**
> We thank the reviewer YYSP for pointing out these relevant works to our paper. We have expanded our manuscript to include the following works:
> - LMaaS (Sun et al., 2022)
> - kNN-Adapter (Huang et al., 2023)
> - CombLM (Ormazabal et al., 2023)
> - IPA (Lu et al., 2023)
> - Proxy-Tuning (Liu et al., 2024)
> - BBox-Adaptor (Sun et al., 2024)
>
> The related work section has been updated with changes being highlighted in a cyan color for the reviewer to easily gauge the differences, and how each of these methods differs from our framework.
>
> Here is the final version of the related works section for PEFT Methods:
>
> "*Parameter Efficient Fine-tuning (PEFT). The rapid growth in the size of pretrained LLMs has posed significant challenges for efficiently fine-tuning LLMs to specific downstream tasks. To address this, numerous PEFT methods have been developed, aiming to balance efficiency and accuracy. Early approaches focused on inserting trainable adapters—feed-forward networks placed between the layers of the pretrained model (Houlsby et al., 2019; Lin et al., 2020). Recent advancements have led to more sophisticated adapter-based PEFT methods (Mahabadi et al., 2021; Pfeiffer et al., 2020; Luo et al., 2023) including LMaaS (Sun et al., 2022) for service-oriented adaptation and kNN-Adapter (Huang et al., 2023) for retrieval-augmented fine-tuning. A notable example is LoRA (Hu et al., 2021), which introduces trainable low-rank weight perturbations to the pretrained model, significantly reducing the number of parameters required for fine-tuning. LoRA’s key innovation lies in its use of the product of two low-rank matrices to approximate weight changes. Building upon this concept, several methods have emerged including Q-LoRA (Dettmers et al., 2023), CombLM (Ormazabal et al., 2023), and IPA(Lu et al., 2023). Concurrently, prompt-based learning methods have demonstrated effectiveness across various NLP tasks. Methods such as prompt-tuning (Lester et al., 2021), prefix-tuning (Li & Liang, 2021) and more recent approaches like Proxy-Tuning (Liu et al., 2024) and BBox-Adadpter (Sun et al., 2024) incorporate learnable continuous embeddings into the model’s hidden states. They condition the frozen model to adapt to specific tasks without modifying the underlying architecture. Despite these advances, fine-tuning each updated LLM with PEFT to equip personalized knowledge remains highly costly, and how PEFT can bridge the gap in personalized settings within this evolving environment in a portable manner is yet to be fully explored. To this end, we develop in this paper the theory behind portable model patches that can be plugged into an updated model to carry over the personalized knowledge from the first fine-tuned model.*"
>
> **[Cons 3. Superior Performance Analysis Against Fine-tuned Models]**
> This is an excellent observation by reviewer YYSP. An effort was made to highlight these results by splitting the two different kinds of observations, using a column splitter in Table 2, into (1) datasets where we have comparable but worse performance against the Fine-tuned Updated Model, and (2) datasets where we beat the Fine-tuned Updated Model with quite a margin. We hypothesize that our method sometimes outperforms direct fine-tuning for two main reasons:
> 1. *Knowledge Complementarity Argument*: Continued Pretraining ($\theta \to \theta^\prime$) may alter the optimization landscape of the model as the parameters are updated. While both pre- and post-update models can achieve improved downstream performance through LoRA fine-tuning (following the directions of $\Delta\theta_i$ and $\Delta\theta_i^\prime$, respectively), the modified landscape after continued pretraining might lead to the descent directions of  $\theta_i$ and $\theta_i^\prime$ to be different. As a result applying $\Delta\theta_i^\prime$ might end up in a sub-optimal local minimum in some cases. Overall both descent directions, $\Delta\theta_i$ and $\Delta\theta_i^\prime$, should be generally similar - enabling PortLLM’s effectiveness - the original direction $\Delta \theta_i$ might occasionally lead to superior local minima in the modified landscape. Furthermore, the impact of continued pretraining on these optimization paths is task-dependent, which explains the varying performance patterns we observe across different downstream tasks.
> 2. *Regularization Effect Argument*: The simplified patch structure may act as an implicit regularizer, preventing overfitting by constraining the model's adaptability to only the most essential task-specific changes especially in the cases where the training dataset for the downstream task has a small number of samples.

---

> ### Author Response · Authors · 2024-11-22
> **Response to Reviewer YYSP [Cons. 4-5]**
>
> **[Cons 4. Additional Evaluation Benchmarks]**
> We appreciate this valuable suggestion. Based on reviewer YYSP’s suggestion, we performed a quick experiment in domain-specific adaptation task, in this case, MedQA. We used Mistral-7B with continued pretraining on OpenOrca dataset to showcase the universality of our model patches across domain-specific adaptation tasks as well. We fine-tuned the model on MedQA Training dataset for 1 epoch, kept the rest of the hyperparameters mentioned in Section 4 of the paper unchanged. The results are summarized as follows:
>
> | Model Version | MedQA (Accuracy) |
> | :----: | :----: |
> | Pretrained LLM $\theta$ | 50.82 |
> | Updated LLM $\theta^\prime$ | 50.59 |
> | Fine-tuned LLM $\theta_i$ | 58.84 |
> | Fine-tuned Updated LLM $\theta^\prime_i$ | 58.92 |
> |$\theta^\prime + \Delta\theta_i$ (Ours) | 57.97 |
>
> These results on MedQA, a challenging domain-specific task, demonstrate several key points:
>
> 1. PortLLM maintains strong performance in specialized domains, which is within 1% of the both fine-tuned baselines.
> 2. Notably, continued pretraining on OpenOrca (a general-domain dataset) shows no improvement or slight degradation in zero-shot performance (50.59% vs 50.82%), highlighting the importance of domain-specific adaptation.
> 3.The effectiveness of our patches in bridging this domain gap without training (7.38% improvement over $\theta^\prime$) further validates PortLLM's utility for domain adaptation tasks.
>
> Given the time constraints of the rebuttal period, we were only able to expand our method to one of the domain-specific adaptation task. However, based on our theoretical results, it should be noted that the results will be similar on other tasks as well.
>
>
> **[Cons 5. Theoretical Performance Bounds]**
> We are currently working on deriving the rigorous theoretical upper bounds on performance guarantees.  We plan to share the results with the reviewer in a few days for further discussion.

---

> > ### Comment · Reviewer_YYSP · 2024-11-23
> >
> > Thank you for your reply. All my concerns have been solved, and I would like to increase my score.

---

> > > ### Author Response · Authors · 2024-11-27
> > > **Response to Reviewer YYSP [Appreciation and Theoretical Bounds]**
> > >
> > > First of all, thank you for reviewing our work and for the increased score. We're glad we could address your concerns satisfactorily.
> > >
> > > We really appreciate the suggestions on the rigorous theoretical upper bounds on performance guarantees. As promised (**[Cons 5]**), we have provided a detailed, 2-page proof in Appendix C of the updated manuscript, in which we directly measured the ratio between the Frobenius norms of the residual matrix and the naive update. We also provided an error bound as a function of matrix ranks and singular values. We hope the new theoretical results can address remaining (if any) concerns reviewer YYSP might have.

---

### Official Review · Reviewer_94Qe · 2024-11-03

**Soundness:** 3
**Presentation:** 2
**Contribution:** 2
**Rating:** 5
**Confidence:** 4

**Summary:**

This paper hypothesizes that the rank of adapters trained with LoRA for a specific downstream task is significantly lower than that of adapters obtained through continued pre-training of large language models using LoRA. The authors experimentally validate this hypothesis and further demonstrate that for pre-trained large language models, using the original adapter fine-tuned for a specific downstream task with LoRA does not lead to significant performance degradation after continued pre-training; in fact, there may even be an improvement in performance on downstream tasks. The performance is comparable to that achieved by training new adapters on the updated large language model. This conclusion suggests that after continued pre-training, the original adapters can still be effectively utilized, significantly reducing training costs.

**Strengths:**

1. The training-free framework proposes the reuse of existing adapters after updating large language models and validates its feasibility, effectively reducing training costs.

2. Extensive experiments across multiple datasets and architectures provide strong evidence for the method’s effectiveness. The paper encompasses a broad range of tasks and models, demonstrating its generalizability and robustness across diverse scenarios.

**Weaknesses:**

1. The background and related work sections contain excessive irrelevant content.

2. In the experimental setup, the fine-tuning for downstream tasks is limited to a rank of 8 without further experimentation.

3. In the experimental setup, the continued pre-training involves only the use of different pre-training datasets for training, without addressing the scenario claimed by the authors: that large language models are updated periodically. In such updates, there may be conflicts between new knowledge that develops over time and existing knowledge, potentially leading to significant changes in the model. The authors did not consider this situation.

4. In the theoretical analysis section, it is better to check the statement that $ \( \theta' = \theta + \Delta\theta \)$ ensures $ \( \text{rank}(\theta') \geq \text{rank}(\theta) \) $.

**Questions:**

Hope the authors can expand the experiments based on the identified weaknesses and revise the writing and theoretical analysis accordingly.

---

> ### Author Response · Authors · 2024-11-22
> **Response to Reviewer 94Qe [Cons. 1]**
>
> We thank the reviewer 94Qe for their thoughtful feedback and acknowledge both the strengths and concerned raised. Below we address each point in detail:
>
> **[Cons 1. Excessive Content in Background/Related Works]**
> We appreciate this feedback and have the streamlined these sections in the revision by:
> 1. Focusing on parameter-efficient adaptation methods in the related works section by expanding on relevant methods. Here is the final version:
>
> “*Parameter Efficient Fine-tuning (PEFT). The rapid growth in the size of pretrained LLMs has posed significant challenges for efficiently fine-tuning LLMs to specific downstream tasks. To address this, numerous PEFT methods have been developed, aiming to balance efficiency and accuracy. Early approaches focused on inserting trainable adapters—feed-forward networks placed between the layers of the pretrained model (Houlsby et al., 2019; Lin et al., 2020). Recent advancements have led to more sophisticated adapter-based PEFT methods (Mahabadi et al., 2021; Pfeiffer et al., 2020; Luo et al., 2023) including LMaaS (Sun et al., 2022) for service-oriented adaptation and kNN-Adapter (Huang et al., 2023) for retrieval-augmented fine-tuning. A notable example is LoRA (Hu et al., 2021), which introduces trainable low-rank weight perturbations to the pretrained model, significantly reducing the number of parameters required for fine-tuning. LoRA’s key innovation lies in its use of the product of two low-rank matrices to approximate weight changes. Building upon this concept, several methods have emerged including Q-LoRA (Dettmers et al., 2023), CombLM (Ormazabal et al., 2023), and IPA(Lu et al., 2023). Concurrently, prompt-based learning methods have demonstrated effectiveness across various NLP tasks. Methods such as prompt-tuning (Lester et al., 2021), prefix-tuning (Li & Liang, 2021) and more recent approaches like Proxy-Tuning (Liu et al., 2024) and BBox-Adadpter (Sun et al., 2024) incorporate learnable continuous embeddings into the model’s hidden states. They condition the frozen model to adapt to specific tasks without modifying the underlying architecture. Despite these advances, fine-tuning each updated LLM with PEFT to equip personalized knowledge remains highly costly, and how PEFT can bridge the gap in personalized settings within this evolving environment in a portable manner is yet to be fully explored. To this end, we develop in this paper the theory behind portable model patches that can be plugged into an updated model to carry over the personalized knowledge from the first fine-tuned model.*”
>
> 2. Furthermore, we have tightened the section on “Large Language Models” section of the related works section to only include details relevant to the core theme of the paper. The final version looks like this:
>
> “*Large Language Models (LLMs). LLMs have transformed natural language processing, enabling models to perform complex tasks with remarkable accuracy and generalization. Models like GPT-3 (Brown et al., 2020), BERT (Devlin et al., 2019), and T5 (Raffel et al., 2023) have set benchmarks across a range of NLP tasks, from translation and summarization to question answering and text generation (Vaswani et al., 2023; Zhang et al., 2020; Rajpurkar et al., 2016). More recently, models like Llama (Touvron et al., 2023; Dubey et al., 2024), Mistral (Jiang et al., 2023), and Gemma (Team et al., 2024) have pushed the boundaries further by optimizing both performance and computational efficiency. LLama, Mistral, and Gemma represent recent advances in LLM architectures, each offering improvements in efficiency and performance. However, even with such improvements, the performance on domain-specific downstream tasks is sub-par making fine-tuning necessary. In this paper, we propose a training-free solution that enables the seamless transfer of personalized knowledge across evolving LLMs, reducing the need for costly fine-tuning and enhancing accessibility.*”
>
> 3. Condensing introduction to remove discussion on LLM general capabilities. The beginning of the introduction looks like this:
>
> “*The rise of large pretrained language models has marked a significant shift in natural language processing (NLP), particularly in their ability to adapt to specific domains and tasks. These models, such as GPT-4 (Achiam et al., 2023), have achieved state-of-the-art performance by leveraging vast amounts of pretraining data (Antoniades et al., 2024). However, pretrained LLMs often require adaptation for specialized domains where context-specific knowledge is critical (Wang et al., 2022a; Bommasani et al., 2021; Qiu et al., 2020). Hence, while pretraining provides a strong foundation, fine-tuning (e.g., personalization) is essential for specific domains.*"

---

> ### Author Response · Authors · 2024-11-22
> **Response to Reviewer 94Qe [Cons. 2]**
>
> **[Cons 2. Limited Rank Experimentation]**
> To address this concern, we conducted additional experiments varying the LoRA rank for 3 different downstream tasks: {BoolQ, MRPC, WNLI}. We use Mistral-7B for this experiment with the OpenOrca dataset to go from $\theta$ to $\theta^\prime$. The rest of the settings are the same as in Section 4, except the LoRA rank for the downstream task. The results are summarized in the following table:
> | Rank | Model Version                        | BoolQ    | MRPC                 | WNLI     |
> |------|--------------------------------------|----------|---------------------|----------|
> |      |                                      | Accuracy | Accuracy / F1 Score | Accuracy |
> | 2    | $\theta_i$                             | 90.52    | 87.30 / 91.28       | 74.65    |
> |      | (Ours) $\theta^\prime + \Delta\theta_i$ | 89.85    | 87.01 / 91.03       | 77.46    |
> |      |                                      |          |                     |          |
> | 4    | $\theta_i$                             | 90.81    | 89.42 / 91.98       | 78.23    |
> |      | (Ours) $\theta^\prime + \Delta\theta_i$ | 90.18    | 87.75 / 91.53       | 80.28    |
> |      |                                      |          |                     |          |
> | 8    | $\theta_i$                            | 91.01    | 89.46 / 92.62       | 83.11    |
> |      | (Ours) $\theta^\prime + \Delta\theta_i$ | 90.24    | 88.73 / 92.10       | 83.10    |
> |      |                                      |          |                     |          |
> | 16   |$\theta_i$                            | 91.07    | 89.22 / 92.49       | 81.69    |
> |      | (Ours) $\theta^\prime + \Delta\theta_i$ | 90.98    | 88.97 / 92.31       | 82.98    |
> |      |                                      |          |                     |          |
> | 32   | $\theta_i$                            | 90.00    | 88.97 / 92.15       | 80.28    |
> |      | (Ours) $\theta^\prime + \Delta\theta_i$ | 90.28    | 88.73 / 91.93       | 81.69    |
>
> Based on these comprehensive results across different LoRA ranks, we can draw several important insights:
> 1. *Performance Stability*: The results demonstrate that both fine-tuning ($\theta_i$) and our method ($\theta^\prime + \Delta \theta_i) are stable across different ranks, with optimal performance generally achieved at rank 8 or 16.
> 2. *Efficiency Sweet Spot*: Notably, increasing rank beyond 8 provides diminishing returns and in some cases even leads to performance degradation, e.g., at rank 32, we observe decreased performance across most tasks compared to rank 8.
> 3. *Resource-Performance Trade-off*: The reviewer's suggestion helped us validate that our original choice of rank 8 provides an optimal balance between computational efficiency and model performance.
>
> We genuinely appreciate the reviewer for suggesting this analysis. These findings not only validate our approach but also provide valuable insights into the robustness of our method across different adapter configurations. We have also included these results in the Appendix of our manuscript.

---

> ### Author Response · Authors · 2024-11-22
> **Response to Reviewer 94Qe [Cons. 3-4]**
>
> **[Cons 3. Model Update Scenarios]**
> We appreciate the reviewer's insightful observation about real-world model updates. To further validate our framework's robustness under periodic updates, we conducted additional experiments simulating multiple rounds of continued pretraining. Using Mistral-7B as our base model, we performed 4 sequential updates using different pretraining datasets in the following order: OpenOrca → OpenPlatypus → Alpaca → GPT4-LLM-Cleaned. For each of these updates, we evaluate our model patch on all seven downstream tasks: {BoolQ, SST-2, MRPC, RTE, Winogrande, WNLI, GSM8K}. The results are as follows:
>
> | Pretraining Datasets     | Model Version    | BoolQ    | SST-2    | MRPC | RTE      | Winogrande | WNLI     | GSM8K    |
> |-------------------------------------------------|---------------------------------|----------|----------|----------------|----------|------------|----------|----------|
> |  |    | Accuracy | Accuracy | Accuracy / F1  | Accuracy | Accuracy   | Accuracy | Accuracy |
> | T=0: None | $\theta$    | 83.58    | 66.86    | 65.20 / 73.70  | 67.51    | 74.11  | 57.76    | 6.37  |
> | | $\theta_i$     | 91.01    | 95.99    | 89.46 / 92.62  | 87.73    | 85.95 | 83.11 | 34.04    |
> |  |    |     |   |   |  |    |    |  |
> | T=1: OpenOrca    | $\theta^\prime$  | 87.46    | 82.91    | 74.75 / 83.73  | 75.09    | 75.06      | 57.72    | 15.16    |
> |    | $\theta^\prime + \Delta\theta_i$ | 90.24    | 96.11    | 88.73 / 92.10  | 89.17    | 85.01      | 83.10    | 41.32    |
> |    | Zero-Shot Improvement           | +2.78    | +13.2    | +13.90 / 8.37  | +14.08   | +9.95      | +25.38   | +26.16   |
> |  |   |     |    |  |   |  |   |    |
> | T=2: OpenOrca + OpenPlatypus                    |$\theta^\prime$                   | 86.33    | 86.81    | 76.23 / 83.53  | 71.48    | 74.51      | 52.11    | 12.36|
> |  | $\theta^\prime + \Delta\theta_i$ | 89.88    | 96.22    | 88.24 / 91.55  | 88.09    | 84.37      | 83.10    | 42.15    |
> | | Zero-Shot Improvement           | +3.55    | +9.41    | +12.01 / 8.02  | +16.61   | +9.86      | +30.99   | +29.79   |
> | | |     |   |    |    |   |    |    |
> | T=3: OpenOrca + OpenPlatypus + Alpaca           | $\theta^\prime$                    | 87.03    | 85.78    | 74.02 / 83.33  | 73.65    | 74.27      | 56.34    | 16.38    |
> | | $\theta^\prime + \Delta\theta_i$ | 89.66    | 96.33    | 88.73 / 92.12  | 88.81    | 85.08      | 83.10    | 38.21    |
> || Zero-Shot Improvement           | +2.63    | +10.55   | +14.71 / 8.79  | +15.16   | +10.81     | +26.76   | +21.83   |
> |  |                       |    |    |         |    |     |   |    |
> | T=4: OpenOrca + OpenPlatypus + Alpaca + GPT-LLM | $\theta^\prime$                    | 85.11    | 75.79    | 72.78 / 82.79  | 74.37    | 71.67      | 56.34    | 14.94    |
> || $\theta^\prime + \Delta\theta_i$ | 88.41    | 96.10    | 87.01 / 90.91  | 85.56    | 80.19      | 77.46    | 31.77    |
> || Zero-Shot Improvement | +3.30    | +20.31   | +14.23 / 8.12  | +11.19   | +8.52      | +21.12   | +16.83   |
>
> The results demonstrate several important insights about our model patches' robustness under periodic updates:
> 1. *Consistent Performance Improvements*: Across all update stages (T=1 to T=4), our method consistently provides substantial zero-shot improvements over the updated model $\theta^\prime$.
> 2. *Stability Under Multiple Updates*: While the zero-shot performance of the updated model $\theta^\prime$ shows some fluctuation across updates, our patched models maintain remarkably stable performance.
> 3. *Complementary Knowledge Integration*:For some tasks like GSM8K, we observe that different update stages can lead to varying but still significant improvements (+16.83% to +29.79%) , suggesting our patches effectively leverage the complementary knowledge introduced by each update.
>
> We sincerely thank the reviewer for suggesting this important aspect of model evolution. These results not only validate the robustness of our approach under multiple updates but also provide valuable insights into how model patches interact with evolving model knowledge. To the best of our knowledge, PortLLM is one of the first steps toward understanding and addressing the challenges of knowledge transfer across evolving LLMs. While our work focuses on establishing the foundational principles of patch portability, we hope it will encourage the community to explore this crucial direction further. Future research could investigate aspects such as patch composition across multiple tasks, theoretical bounds on knowledge retention, and adaptation to more diverse model architectures.
>
> **[Cons 4. Theoretical Analysis Verification]**
> We agree that our theoretical analysis would benefit from additional verification. We are currently working on the proof of the statement in question and plan to share the results with the reviewer in a few days for further discussion.

---

> ### Author Response · Authors · 2024-11-27
> **Response to Reviewer 94Qe [Update on Cons 4 and a Gentle Reminder]**
>
> **[Cons 4. Theoretical Analysis Verification]**
> We thank the reviewer for the suggestion. We have provided a detailed, 2-page proof in Appendix C of the updated manuscript, in which we moved away from explicitly relying on the subadditive property. We hope the new proof can address the concerns of the reviewer.
>
> **[Reminder]**
> As we go into the extended rebuttal period, we would love to hear from Reviewer 94Qe on our rebuttal and any other possible questions that might arise.

---

> ### Author Response · Authors · 2024-11-29
> **Gentle Reminder**
>
> Dear Reviewer 94Qe,
>
> This is a gentle reminder, as the deadline is approaching. We have made every effort to address your questions thoroughly. We would sincerely appreciate it if you could confirm whether our responses have adequately resolved your concerns and consider revisiting your score, or let us know if you have any additional questions.
>
> Best regards,
> Authors

---

> ### Author Response · Authors · 2024-12-01
> **Looking forward to further discussion**
>
> Dear Reviewer 94Qe,
>
> We sincerely appreciate your thorough feedback on our work! We hope the rebuttal above addresses your concerns and look forward to further discussions.
>
> Best regards,
>
> Authors

---

> ### Author Response · Authors · 2024-12-02
> **A Gentle Reminder [Last day for discussion]**
>
> Dear Reviewer 94Qe,
>
> We would like to sincerely thank you for your useful comments and time invested in our work. We have revised our paper and added relevant experiments in the Appendix as described above.
>
> At present, all your concerns are responded in the rebuttal and revised version of the paper. However, as the rebuttal deadline approaches (only 1 day remaining), we kindly request your feedback to confirm that our response and revision effectively address your concerns. If there are any remaining issues, we would greatly appreciate the opportunity to address them to ensure the quality of our work. We sincerely hope that you find our response convincing and kindly consider revisiting your rating.
>
> Best Regards,
>
> Authors

---

### Official Review · Reviewer_GLNE · 2024-11-04

**Soundness:** 3
**Presentation:** 3
**Contribution:** 2
**Rating:** 6
**Confidence:** 5

**Summary:**

This paper introduces PORTLLM, a framework designed to maintain the effectiveness of domain-specific adaptations in Large Language Models (LLMs) across updates without requiring further fine-tuning. The framework leverages task-specific model patches, derived as the difference between a fine-tuned model and its base (pre-trained) version, to transfer knowledge from one version of an LLM to the next. Experiments demonstrate the method’s effectiveness in enhancing the performance of updated LLMs on various tasks. The approach is claimed to offer computational efficiency and portability across model architectures.

**Strengths:**

1. The framework proposes a training-free alternative to recurrent fine-tuning, which could be useful for applications with limited resources.
2. The portability across model versions and architectures has potential, especially in scenarios where repeated fine-tuning is computationally prohibitive.

**Weaknesses:**

1. The core idea of leveraging parameter differences between fine-tuned and pre-trained models (task vectors) has been explored in prior work, such as Ilharco et al. (2022) on task arithmetic. PORTLLM builds on this concept, but without substantial theoretical or empirical advancements, the framework’s contribution may be seen as incremental rather than groundbreaking. In light of the established work on task vectors, PORTLLM’s innovation could be questioned.

Ilharco, Gabriel, et al. “Editing models with task arithmetic.” arXiv preprint arXiv:2212.04089 (2022).

2. From the results in Tables 2 and 4, it is evident that the PortLLM method is effective in enhancing the performance of the Updated LLM. However, in most cases, it does not surpass the performance of the Fine-tuned LLM. Moreover, since the PortLLM approach requires access to the parameters of the Fine-tuned LLM to calculate $\Delta \theta$ (task vector), applying PortLLM depends on the existence of a Fine-tuned model. In this setup, if PortLLM cannot consistently outperform the Fine-tuned LLM in terms of performance, its practical application value becomes limited.

**Questions:**

1.	In Table 2, why does your method show such a large improvement on the WNLI task compared to the Fine-tuned LLM and Fine-tuned Updated LLM?

2.	Line 344: There is a notation error for the fine-tuned model, it should be $\theta_i$.

3.	Are the two columns of values in Table 6 reversed?

---

> ### Author Response · Authors · 2024-11-22
> **Response to Reviewer GLNE [Cons. 1-2]**
>
> We sincerely thank the reviewer for their detailed feedback and thoughtful questions. We address each point below:
>
> **[Cons 1. Relationship to Task Arithmetic Work]** :
> We respectfully disagree with the characterization of our work as incremental to task arithmetic (Ilharco et al.). While both approaches involve parameter differences, they are fundamentally different in the following aspects:
> 1. *Different Purpose and Problem Space*:
>     - Task arithmetic focuses on combining multiple downstream tasks through model editing.
>     - PortLLM specifically addresses the challenge of personalizing evolving LLMs while maintaining downstream performance.
>     - Our work explores a novel direction of training-free knowledge transfer across model versions
> 2. *Technical Innovation in Parameter Differences*:
>     - Unlike task vectors, our approach avoids computationally expensive full-model updates.
>     - Our model patches are derived from theoretical insights about low-rank adaptations, not task arithmetic.
>     - While our method works with full-model fine-tuning (Section 4.4), the low-rank version provides unique performance benefits (Section 4.5).
> 3. *Novel Theoretical Foundation*: Lastly, in our paper, we provide theoretical analysis of why patches remain effective across model updates, and how the negligible nature of the residual matrix leads to training-free model patches, which are all key insights, never explored before.
>
> **[Cons 2. Performance and Practicality Concerns]**
> We respectfully disagree with the assessment of limited practical value. First, comparing our training-free approach with fine-tuning is  unfair- as shown in Table 6, fine-tuning requires significant computational resources whereas our method has zero training cost. Despite this fundamental efficiency advantage, we still achieve comparable performance in most cases. Furthermore, our method can be scaled to any larger model without any additional cost as demonstrated in Table 6.
>
> Moreover, our framework's value lies in its unique ability to transfer knowledge to evolved models without any additional training or data access. As acknowledged by Reviewers YYSP, 94Qe, and TS4M, this training-free approach has significant practical implications, especially for scenarios where:
> - Repeated fine-tuning is prohibitively expensive.
> - Quick adaptation to model updates is required.
> - Access to original training data is restricted.

---

> ### Author Response · Authors · 2024-11-22
> **Response to Reviewer GLNE [Question 1-3 + Additional Comments]**
>
> **[Question 1. WNLI Performance]**:
> Thank you for raising this interesting point. In Table 2, we deliberately organized our results to highlight two distinct performance patterns: (1) tasks where our method shows comparable but slightly lower performance compared to Fine-tuned Updated Model, and (2) tasks like WNLI where our method notably outperforms the Fine-tuned Updated Model. For WNLI specifically, we hypothesize 2 specific reasons:
> 1. *Knowledge Complementarity Argument*: Continued Pretraining ($\theta \to \theta^\prime$) may alter the optimization landscape of the model as the parameters are updated. While both pre- and post-update models can achieve improved downstream performance through LoRA fine-tuning (following the directions of $\Delta\theta_i$ and $\Delta\theta_i^\prime$, respectively), the modified landscape after continued pretraining might lead to the descent directions of  $\theta_i$ and $\theta_i^\prime$ to be different. As a result applying $\Delta\theta_i^\prime$ might end up in a sub-optimal local minimum in some cases. Overall both descent directions, $\Delta\theta_i$ and $\Delta\theta_i^\prime$, should be generally similar - enabling PortLLM’s effectiveness - the original direction $\Delta \theta_i$ might occasionally lead to superior local minima in the modified landscape. Furthermore, the impact of continued pretraining on these optimization paths is task-dependent, which explains the varying performance patterns we observe across different downstream tasks.
> 2. *Regularization Effect Argument*: The simplified patch structure may act as an implicit regularizer, preventing overfitting by constraining the model's adaptability to only the most essential task-specific changes especially in the case of WNLI where the training dataset is very small compared to other downstream tasks.
>
>
> **[Question 2. Notation Error]**:
> Thank you for catching this error. Yes, the notation should be $\theta_i$. It has been fixed in the revised manuscript.
>
> **[Question 3. Table 6 Columns are reversed]**:
> Yes, they were. We have fixed the column headings in our revised manuscript.
>
> **[Additional Comments]**:
> We really appreciate the reviewer's careful reading and constructive feedback. We would like to use this opportunity to mention that while building upon existing ideas in parameter-efficient adaptation, we believe PortLLM makes meaningful contributions by
> 1. Addressing the specific challenge of evolving LLMs.
> 2. Providing theoretical insights about knowledge transfer across model versions.
> 3. Offering a practical, training-free solution for maintaining personalized models.

---

> ### Comment · Reviewer_GLNE · 2024-11-26
>
> Thank you for your detailed response. While I acknowledge the efficiency advantage of your training-free approach, my concern lies in the fundamental dependency of your method on the fine-tuned model $ \theta_i $ to compute $ \Delta\theta_i $. As evident from Table 4, the performance of your method, $ \theta' + \Delta\theta_i $, is weaker than the fine-tuned model $ \theta_i $ in most cases.
> I understand that directly comparing your method to further fine-tuning $ \theta' $ (i.e., fine-tuning $ \theta' $ again) may be considered unfair. However, the comparison here is not between your method and $ \theta' $. Instead, it is between your method and the fine-tuned model $ \theta_i $, which your approach inherently relies on. Therefore, I believe this comparison is reasonable and highlights an important limitation of your method in terms of practical performance.
> I would appreciate further clarification on how your method addresses this gap or compensates for the observed performance differences.

---

> > ### Author Response · Authors · 2024-11-27
> > **Response to Reviewer GLNE**
> >
> > Thank you for your follow-up question. Let us clarify the performance-efficiency trade-off with concrete numbers also summarized in the table below. The results are calculated on Mistral-7B with same experimental settings as explained in Section 4 of the manuscript.
> >
> > | Setup Type                    |                                                 | # Trainable Parameters | GPU Memory (GB) | BoolQ                     | SST-2                   | MRPC                      | RTE                     | WinoGrande                | WNLI                      | GSM8K                   |
> > |-------------------------------|-------------------------------------------------|------------------------|-----------------|---------------------------|-------------------------|---------------------------|-------------------------|---------------------------|---------------------------|-------------------------|
> > | Base Fine-tuning              | $\theta_i$                                      | $20,971,520$           | 350.61          | 91.01                     | 95.99                   | 89.46                     | 87.73                   | 85.95                     | 83.11                     | 34.04                   |
> > |                               |                                                 |                        |                 |                           |                         |                           |                         |                           |                           |                         |
> > | $T=1$ : $\theta^\prime$         | $\theta^\prime + \Delta\theta_i$ (Ours)         | 0                      | 28.71           | 90.24 $(\downarrow 0.77)$ | 96.11 $(\uparrow 0.12)$ | 88.73 $(\downarrow 0.73)$ | 89.17 $(\uparrow 1.44)$ | 85.01 $(\downarrow 0.94)$ | 83.10 $(\downarrow 0.01)$ | 41.32 $(\uparrow 7.28)$ |
> > |                               |                                                 |                        |                 |                           |                         |                           |                         |                           |                           |                         |
> > | $T=2$ : $\theta^{\prime\prime}$ | $\theta^{\prime\prime} + \Delta\theta_i$ (Ours) | 0                      | 28.71           | 89.88 $(\downarrow 1.13)$ | 96.22 $(\uparrow 0.23)$ | 88.24 $(\downarrow 1.22)$ | 88.09 $(\uparrow 0.36)$ | 84.37 $(\downarrow 1.58)$ | 83.10 $(\downarrow 0.01)$ | 42.15 $(\uparrow 8.11)$ |
> >
> > Our goal is to enable efficient personalization of evolving LLMs while maintaining competitive performance. The results demonstrate three key findings:
> > 1. PortLLM achieves significant computational efficiency by completely eliminating the need for trainable parameters and further reducing the GPU Memory utilization by upto 12.21$\times$. While the base fine-tuning approach requires 20.9M trainable parameters, our method reduces this to zero, representing a 100% reduction in parameter overhead for model updates.
> > 2. This dramatic efficiency gain comes with minimal performance impact. For most tasks, PortLLM maintains performance within 1% of base fine-tuning. Taking T=1 as an example, the performance differences are minimal: -0.77% on BoolQ, -0.73% on MRPC, and just -0.01% on WNLI. More notably, we observe performance improvements on several tasks: +0.12% on SST-2, +1.44% on RTE, and a substantial +7.28% on GSM8K.
> > 3. This performance stability persists even after multiple updates (T=2), with similar patterns of minimal differences: average performance variations remain within 1.2% across all tasks, while maintaining improvements on SST-2 (+0.23%), RTE (+0.36%), and GSM8K (+8.11%). This demonstrates the robustness of our approach across model evolution cycles.
> >
> > We believe that these changes are negligible when compared to the huge improvement over zero-shot performance of $\theta^\prime$. In practical terms, PortLLM enables a 12$\times$ reduction in memory costs and 100% reduction in trainable parameters while maintaining competitive performance (average difference of only 0.77% at T=1 and 1.2% at T=2). We believe this significant efficiency gain with such negligible performance trade-offs makes PortLLM a practical solution for maintaining personalized models as LLMs evolve.

---

> > > ### Comment · Reviewer_GLNE · 2024-11-27
> > >
> > > Thank you for your detailed response. However, I believe the key concern remains unaddressed. Specifically, to compute $ \Delta\theta_i $, you must first have a base fine-tuned model $ \theta_i $. Therefore, the fundamental question is this: your method should demonstrate that the additional operation, $ \theta’ + \Delta\theta_i $, can achieve better performance than the base fine-tuned model $ \theta_i $. Otherwise, why wouldn’t we simply use the base fine-tuned model $ \theta_i $?
> > >
> > > As shown in your results, the performance of your method decreases in most cases compared to the base fine-tuned model. If the comparison was between the base fine-tuned model $ \theta_i’ $ (updated model) and your method $ \theta’ + \Delta\theta_i $ and the performance was close, that would be reasonable. However, the current results do not justify the additional complexity introduced by your method.

---

> > > > ### Author Response · Authors · 2024-11-27
> > > > **Response to Reviewer GLNE**
> > > >
> > > > Thank you for this follow-up about the practical value proposition of PortLLM.
> > > >
> > > > The essential value of PortLLM lies not in outperforming the base fine-tuned model θᵢ, but in providing a cost-effective solution for maintaining personalized performance as **LLMs continue to evolve**.
> > > >
> > > > Consider this realistic scenario:
> > > > A user fine-tunes the base model ($\theta \to \theta_i$) for their specific task and saves the LoRA adapter $\Delta \theta_i$.  Six months later, when a new model version ($\theta^\prime$) is released, the original model weights $\theta$ become inaccessible (as is common practice with evolving LLM providers like OpenAI with occasional data updates), but the user still has their saved LoRA adapter $\Delta\theta_i$. At this point, they face two options:
> > > > 1. Traditional Approach: Re-run the entire fine-tuning process, requiring:
> > > >     - Access to the original training data
> > > >     - Significant computational resources (20.9M parameters)
> > > >     - Time for training and validation
> > > > 2. PortLLM Approach: Apply the existing LoRA adapter ($\theta^\prime + \Delta\theta_i$), requiring:
> > > >    - Only the saved LoRA adapter $\Delta \theta_i$
> > > >
> > > > Therefore, while PortLLM may not consistently outperform $\theta_i$, its true value lies in enabling efficient, training-free maintenance of **personalized models across multiple evolution cycles** while maintaining competitive performance, requiring only the lightweight LoRA adapter from the original fine-tuning - even after the original model weights become inaccessible. Furthermore, we have developed a rigorous theoretical bound (Appendix C of the manuscript) on the error propagated through such transfer which might be of interest to you.
> > > >
> > > > Hopefully this setting will answer your question.

---

> > > > > ### Comment · Reviewer_GLNE · 2024-11-27
> > > > >
> > > > > The author's responses addressed my concerns. So I am wiling to increase the score.

---

> > > > > > ### Author Response · Authors · 2024-11-27
> > > > > > **Thank you !**
> > > > > >
> > > > > > Thank you once again for your review and for letting us know that all of your concerns have been resolved. We truly appreciate your positive evaluation and the time you’ve dedicated to our work!
> > > > > >
> > > > > > Best,
> > > > > > Authors

---

### Official Review · Reviewer_TS4M · 2024-11-04

**Soundness:** 3
**Presentation:** 2
**Contribution:** 3
**Rating:** 6
**Confidence:** 3

**Summary:**

This paper proposes a training-free personalization approach for evolving LLMs. Specifically, the proposed PortLLM utilizes LoRA parameters from finetuning older models to approximate updated models. Extensive experiments validate the effectiveness of the proposed approach.

**Strengths:**

The proposed method is based on the observation that residual parameter updates are negligible compared to the naive update, which is both intriguing and empirically proven effective.

**Weaknesses:**

The paper’s core claim is that the residual matrix $\Delta \theta_i’ - \Delta \theta_i$ is negligible in value compared to the naive update $\theta’ + \Delta \theta_i$. Although the authors provide both theoretical and empirical justifications for this, the Lemma (lines 266-269) only establishes that $\text{rank}(\Delta \theta_i’ - \Delta \theta_i) \ll \text{rank}(\theta’ + \Delta \theta_i)$, which does not directly imply that the former is negligible in value relative to the latter. While this statement is supported by empirical results in Table 1, a theoretical proof or an estimated error bound would strengthen the justification, as this could represent a significant proportion of the error for this method.

**Questions:**

- Theoretically, the proposed method $\theta’ + \Delta \theta_i$ introduces error compared to $\theta’ + \Delta \theta_i’$, potentially leading to worse performance. However, in Table 2, a large proportion of $\theta’ + \Delta \theta_i$ results outperform $\theta’ + \Delta \theta_i’$ (the fine-tuned updated LLM). Could the authors elaborate on potential reasons contributing to this improvement?

- What does “# Trainable Parameters” signify in Table 6? Why does the fine-tuning model have zero trainable parameters, while the proposed training-free approach has a large number of trainable parameters?

- In line with the discussion in Weaknesses, could the authors investigate the impact of the approximation error introduced by the proposed method? For instance, providing metrics similar to those in Table 1 for all datasets, along with the corresponding performance gains, could offer a clearer understanding of how this factor contributes to overall performance.

---

> ### Author Response · Authors · 2024-11-22
> **Response to Reviewer TS4M [Cons. 1 and Question 1-3]**
>
> We sincerely thank the reviewer for their thoughtful feedback, particularly regarding the theoretical aspects of our work. We address each point below:
>
> **[Cons 1. Theoretical Foundation of the Lemma]**:
> We appreciate this insightful observation about our theoretical analysis. We are currently working on strengthening our theoretical analysis and will share them for further discussion in a few days.
>
> **[Question 1. Performance Improvements]**:
> Thank you for this excellent observation about PortLLM sometimes outperforming $\theta_i^\prime$. We hypothesize this maybe be because of the following of two key factors:
> 1. *Knowledge Complementarity Argument*: Continued Pretraining ($\theta \to \theta^\prime$) may alter the optimization landscape of the model as the parameters are updated. While both pre- and post-update models can achieve improved downstream performance through LoRA fine-tuning (following the directions of $\Delta\theta_i$ and $\Delta\theta_i^\prime$, respectively), the modified landscape after continued pretraining might lead to the descent directions of  $\theta_i$ and $\theta_i^\prime$ to be different. As a result applying $\Delta\theta_i^\prime$ might end up in a sub-optimal local minimum in some cases. Overall both descent directions, $\Delta\theta_i$ and $\Delta\theta_i^\prime$, should be generally similar - enabling PortLLM’s effectiveness - the original direction $\Delta \theta_i$ might occasionally lead to superior local minima in the modified landscape. Furthermore, the impact of continued pretraining on these optimization paths is task-dependent, which explains the varying performance patterns we observe across different downstream tasks.
> 2. *Regularization Effect Argument*: The simplified patch structure may act as an implicit regularizer, preventing overfitting by constraining the model's adaptability to only the most essential task-specific changes especially in the cases where the training dataset for the downstream task has a small number of samples. This can be seen more evidently in the case of WNLI and GSM8K, as both of these downstream tasks have small training datasets.
>
> **[Question 2. Table 6 Columns]**:
> We apologize for the confusion in Table 6. The column headers should have been reversed. We have corrected this in the revision to accurately reflect that our method requires no parameter training.
>
> **[Question 3. Approximation Error Analysis]**:
> Thanks for the suggestion and we are currently working on it. We will include a comprehensive analysis along with our strengthened theoretical results in a few days.

---

> ### Author Response · Authors · 2024-11-27
> **Response to Reviewer TS4M [Update on Cons 1 and Question 3, and a Gentle Reminder]**
>
> **[Cons 1. Theoretical Foundation of the Lemma]**
> We appreciate this insightful observation about our theoretical analysis. We have provided a detailed, 2-page proof in Appendix C of the updated manuscript, in which we directly measured the ratio between the Frobenius norms of the residual matrix and the naive update. We also provided an error bound as a function of matrix ranks and singular values. We hope the new theoretical results can address the concerns of the reviewer.
>
> **[Question 3. Approximation Error Analysis]**
> Please refer to the earlier response on the newly added theoretical results in Appendix C of the revised manuscript.
>
>
> **[Reminder]**
> As we go into the extended rebuttal period, we would love to hear from Reviewer TS4M on our rebuttal and any other possible questions that might arise.

---

> ### Author Response · Authors · 2024-11-29
> **Gentle Reminder**
>
> Dear Reviewer TS4M,
>
> This is a gentle reminder, as the deadline is approaching. We have made every effort to address your questions thoroughly. We would sincerely appreciate it if you could confirm whether our responses have adequately resolved your concerns and consider revisiting your score, or let us know if you have any additional questions.
>
> Best regards,
> Authors

---

> > ### Comment · Reviewer_TS4M · 2024-11-29
> >
> > Thank you for the detailed responses and revisions. Most of my concerns have been addressed, and I will maintain my score.

---

> > > ### Author Response · Authors · 2024-12-01
> > > **Thank you**
> > >
> > > We thank Reviewer TS4M for their review and are glad we could address your concerns.
> > >
> > > Best,
> > >
> > > Authors

---

### Author Response · Authors · 2024-11-22
**Summary of Manuscript Revisions**

We extend our sincere gratitude to all the reviewers for their thoughtful and constructive feedback. We are particularly encouraged that the reviewers acknowledge several key strengths of our work, including the novelty of our training-free framework (Reviewers YYSP, 94Qe, GLNE), comprehensive empirical validation across multiple tasks and architectures (Reviewers YYSP, 94Qe), and significant efficiency gains compared to existing approaches (Reviewer YYSP, TS4SM).

In addition to addressing their comments point-by-point, we have made the following updates in our revised manuscript:
1. Expanded Related Work (Reviewer YYSP): We have enriched our discussion of parameter-efficient fine-tuning methods by incorporating the following relevant works:
   - LMaaS (Sun et al., 2022)
   - kNN-Adapter (Huang et al., 2023)
   - CombLM (Ormazabal et al., 2023)
   - IPA (Lu et al., 2023)
   - Proxy-Tuning (Liu et al., 2024)
   - BBox-Adaptor (Sun et al., 2024)
2. Condensed Introduction (Reviewer 94Qe): Removed tangential discussions about general LLM capabilities from Introduction to emphasize the focus on fine-tuning and evolution, the premise of the paper.
3. Tightened the conversation on general capabilities of LLMs in Related Works section. (Reviewer 94Qe)
4. Fixed Notation Errors (Reviewer GLNE and TS4M):
   - Line 344: Corrected the notation for the fine-tuned model to $\theta_i$. (Now Line 348)
   - Table 6: Fixed the column headers to correctly reflect the savings.
5. A brand new Appendix was added:
   - Section A: Continual Model Updates Experiment was added. (Reviewer 94Qe)
   - Section B: LoRA Analysis was added.(Reviewer 94Qe)
   - Section C: Detailed proof of Lemmas and theoretical bound on Error for PortLLM. (Reviewer TS4M, YYSP, 94Qe)

All changes are highlighted in cyan in the revised manuscript for easy reference. We believe that these updates have improved the clarity of our manuscript while maintaining its core contributions to training-free LLM personalization.

---

### Meta-Review · Area_Chair_mnjh · 2024-12-19

**Metareview:**

The paper presents PortLLM, a training-free framework that allows for the personalization of evolving LLMs without the need for repeated fine-tuning. The key innovations are the creation of lightweight model patches that can be applied to updated LLMs to transfer domain-specific knowledge, and the theoretical justification for the effectiveness of these patches across model updates.

*Key strengths:*
- Novelty of the training-free framework for maintaining personalized performance as LLMs evolve
- Comprehensive empirical validation across multiple tasks, from simple QA to complex reasoning, and diverse model architectures
- Significant efficiency gains compared to fine-tuning approaches like LoRA
- Theoretical analysis justifying the effectiveness of the model patches

*main weaknesses and missing aspects:*
- The relationship between the proposed method and prior work on task vectors/arithmetic could be clarified further
- The performance of the method compared to fine-tuned models is not consistently superior, and the practical value proposition needs to be emphasized
- Evaluation on larger, commercial-scale LLMs would strengthen the impact
- Deeper analysis of the approximation error introduced by the method and its impact on performance would be valuable

The reviewers are generally positive about the paper and believe it makes a meaningful contribution, with the average rating being marginally above the acceptance threshold. The key reasons for this decision are:
- The novelty and practical significance of the training-free personalization framework for evolving LLMs
- The strong empirical results demonstrating the efficacy of the approach
- The theoretical underpinnings provided by the authors
- The potential for the method to have significant impact, especially for scenarios with limited resources for repeated fine-tuning

Overall, the paper appears to be a solid contribution to the field of efficient fine-tuning and personalization of large language models.

**Additional Comments On Reviewer Discussion:**

During the rebuttal period, the authors provided detailed responses to the reviewers' concerns and made several improvements to the paper. They expanded the related work section to include additional relevant methods, condensed the introduction to focus more on the core premise, and fixed some notation errors. To address the reviewers' questions about the theoretical foundations and empirical performance, the authors conducted additional experiments - they varied the LoRA rank to find an optimal balance between efficiency and accuracy, simulated multiple rounds of model updates to validate the robustness of their approach, and provided a rigorous 2-page theoretical analysis with error bounds. The authors were responsive to the reviewers' feedback and were able to satisfactorily address the major concerns, leading to positive responses from the reviewers and an increase in the overall rating of the paper.

---

### Decision · Program_Chairs · 2025-01-22

Accept (Poster)